

**Global Soil Consumption of Atmospheric Carbon Monoxide:**
**An Analysis Using a Process-Based Biogeochemistry Model**
Licheng Liu[1], Qianlai Zhuang[1,2], Qing Zhu[1,3], Shaoqing Liu[1,4], Hella van Asperen[5], Mari
Pihlatie[6,7]

[1]Department of Earth, Atmospheric, Planetary Sciences, Purdue University, West Lafayette, IN 47907,

USA

[2]Department of Agronomy, Purdue University, West Lafayette, IN 47907, USA

[3]Climate Sciences Department, Climate & Ecosystem Sciences Division, Lawrence Berkeley National

Laboratory, Berkeley, CA 94720, USA

[4]Department of Earth Sciences, University of Minnesota, Minneapolis, MN, 55455,USA

[5]Institute of Environmental Physics, University of Bremen, Otto-Hahn-Allee 1, Bremen, 28359, Germany

[6]Department of Physics, University of Helsinki, P.O. Box 48, 00014 University of Helsinki, Finland

[7]Department of Forest Sciences, P.O. Box27, 00014 University of Helsinki, Finland

*Correspondence to:* Qianlai Zhuang(qzhuang@purdue.edu)





**Abstract:** Carbon monoxide (CO) plays an important role in controlling the oxidizing capacity of the atmosphere by reacting with OH radicals that affect atmospheric methane ($CH_4$) dynamics. We develop a process-based biogeochemistry model to quantify CO exchange between the soil and the atmosphere at the global scale. The model is parameterized using CO flux data from the field and laboratory experiments for eleven representative ecosystem types. The model is then extrapolated to the global terrestrial ecosystems. Global soil gross consumption, gross production, and net flux of the atmospheric CO are estimated to be 132-154, 29-36 and 102-119 Tg CO $yr^{-1}$ (1Tg = $10^{12}$ g), respectively, assuming a constant spatially distributed atmospheric CO concentration (~128 ppbv) during the 20th century. When satellite-based atmospheric CO concentration data are used, our estimates of the soil gross consumption are 180-197 Tg CO $yr^{-1}$ in the period of 2000-2013. Tropical evergreen forest, savanna and deciduous forest areas are the largest sinks at 93 Tg CO $yr^{-1}$. Soil CO gross consumption is sensitive to air temperature and atmospheric CO concentration while gross production is sensitive to soil organic carbon (SOC) stock and air temperature. Under future climate scenarios, the soil gross consumption, gross production and net flux of CO will increase at 0.15-1.23, 0.04-0.3 and 0.12-0.94 Tg CO $yr^{-2}$ during 2014-2100, reaching 162-194, 36-44, and 126-150 Tg CO $yr^{-1}$ by the end of the 21st century, respectively. Areas near the equator, Eastern US, Europe and eastern Asia will be the largest sinks due to optimum soil moisture and high temperature. The annual global soil net flux of atmospheric CO is primarily controlled by air temperature, soil temperature, SOC and atmospheric CO concentrations, while its monthly variation mainly determined by air temperature, precipitation, soil temperature and soil moisture. Our process-based soil CO dynamics model and analysis shall benefit the modeling of the global climate and atmospheric chemistry.

## 1. Introduction

Carbon monoxide (CO) plays an important role in controlling the oxidizing capacity of the atmosphere by reacting with OH radicals (Logan et al., 1981; Crutzen, 1987; Khalil & Rasmussen, 1990; Prather et al., 1995; Prather & Ehhalt, 2001). CO in the atmosphere can directly and indirectly influence the fate of critical greenhouse





gases such as methane ($CH_4$) and ozone ($O_3$) (Logan et al., 1981; Crutzen & Gidel,
1983; Guthrie, 1989; Khalil & Rasmussen, 1990; Lu & Khalil, 1993; Daniel & Solomon,
1998; Prather & Ehhalt, 2001; Tan and Zhuang, 2012). Although CO itself absorbs only
a limited amount of infrared radiation from the Earth, the cumulative indirect radiative
forcing of CO may be even larger than that of the third powerful greenhouse gas, nitrous
oxide ($N_2O$, Myhre et al., 2013). Current estimates of global CO emissions from both
anthropogenic and natural sources range from 1550 to 2900 Tg CO $yr^{-1}$, which are
mainly from anthropogenic and natural direct emissions and from the oxidation of
methane and other Volatile Organic Compounds (VOC) (Prather et al., 1995; Khalil et
al., 1999; Bergamaschi et al., 2000; Prather & Ehhalt, 2001, Stein et al., 2014).
Chemical consumption of CO by atmospheric OH and the biological consumption of CO
by soil microbes are two major sinks of the atmospheric CO (Conrad, 1988; Lu & Khalil,
1993; Prather et al., 1995; Prather & Ehhalt, 2001; Yonemura et al., 2000; Whalen &
Reeburgh, 2001).

Soils are globally considered as a major sink for CO due to microbial activities

(Conrad and Seiler, 1982; Potter et al., 1996; Whalen and Reeburgh, 2001; King and
Weber, 2007). A diverse group of soil microbes including carboxydotrophs,
methanotrophs and nitrifiers are capable of oxidizing CO (Ferenci et al., 1975; Jones
and Morita, 1983; Bender and Conrad, 1994; King and Weber, 2007). Annually, 10-25%
of CO emissions were consumed by soils (Sanhueza et al., 1998; Khalil et al., 1999;
King, 1999a; Bergamaschi et al., 2000; Prather & Ehhalt, 2001; Chan & Steudler, 2006).
Potter et al. (1996) reported the global soil consumption be 16-50 Tg CO $yr^{-1}$, by using a
single box model approach over the upper 5 cm of soils. Other estimates showed large
ranges using simple assumptions, such as 115–230 Tg CO $yr^{-1}$ based on a constant
dry deposition velocity (the uptake rate divided by the CO concentration) of 0.03 cm $s^{-1}$
(Sanhueza et al., 1998); 300 Tg CO $yr^{-1}$ using the same constant deposition velocity
and zero deposition velocity value in deserts and areas with monthly mean
temperatures below 0 °C with different approaches (Bergamaschi et al., 2000); 190-580
Tg CO $yr^{-1}$ using empirical approaches with a higher probability for lower values (King,
1999). Besides, reported CO dry deposition velocities (0 to 0.004m $s^{-1}$) for vegetated
surfaces based on measurements are relatively low compared with other substances





(King, 1999a; Castellanos et al., 2011). To date, there are still large uncertainties in
estimating soil CO consumption, ranging from 15 to 640 Tg CO yr$^{-1}$. Although soil CO
consumption and its environmental controls have been heavily studied, the impacts of
long-term changes in climate and human activities on the atmosphere-biosphere CO
exchange are still not clear (King & Weber, 2007; Vreman et al., 2011; He and He, 2014;
Pihlatie et al., 2016). Moreover, production of CO has been widely found in soils, plant
roots, living and degrading plant materials and degrading organic matter (Pihlatie et al.,
2016). CO production is dominantly due to abiotic processes such as thermal- and
photo-degradation of organic matter or plant material (Conrad and Seiler, 1985b; Tarr et
al., 1995; Schade et al., 1999; Derendorp et al., 2011; Lee et al., 2012; van Asperen et
al., 2015; Fraser et al., 2015, Pihlatie et al., 2016), except for a few cases of anaerobic
formations. Photo-degradation includes direct photo-degradation due to absorbing
radiation by light-absorbing molecules and indirect photo-degradation due to radiation
energy transferring to non-light-absorbing molecules (King et al., 2012). Thermal-
degradation is identified as the temperature-dependent degradation of carbon in the
absence of radiation and possibly oxygen (Derendorp et al., 2011; Lee et al., 2012; van
Asperen et al., 2015; Pihlatie et al., 2016). Previous field and laboratory studies on the
role of direct or indirect abiotic degradation showed very contrasting results, primarily
due to the challenge of separation between CO formation through thermal-degradation
and photo-degradation, because they can both occur simultaneously and the indirect
photo-degradation may occur even without solar radiation if thermal energy is suitable
(Lee et al., 2012).

Little focus has been placed so far on the role of net CO budget (including soil

CO consumption and production) in global climate modeling. Most top-down models
apply a dry deposition scheme based on the resistance model of Wesely (1989). Such
schemes give a wide range of dry deposition velocities (Stevenson et al., 2006). Only a
few models (MOZART-4, Emmons et al., 2010; CAM-chem, Lamarque et al., 2012)
have extended their dry deposition schemes with a parameterization for CO and H$_2$
uptake by oxidation from soil bacteria and microbes following the work of Sanderson et
al. (2003), which itself was based on extensive measurements from Yonemura et al.
(2000). Potter et al. (1996) developed a bottom-up model to simulate CO consumption





and production at the global scale. This model is a single box model, only considers top
5cm depth of soil and does not have explicit microbial factors, which might have
underestimated CO consumption (Potter et al., 1996; King, 1999a). Current bottom-up
CO modeling approaches are mostly based on a limited number of CO *in situ*
observations or laboratory studies to quantify regional and global soil consumption
(Potter et al., 1996; Sanhueza et al., 1998; Khalil et al., 1999; King, 1999a;
Bergamaschi et al., 2000; Prather & Ehhalt, 2001). To our knowledge, no detailed
process-based model of soil-atmospheric exchange of CO has been published in the
recent 15 years. One reason is an incomplete understanding of biological processes of
CO emission and uptake (King & Weber, 2007; Vreman et al., 2011; He and He, 2014;
Pihlatie et al., 2016). Another reason is a lack of long-term CO flux measurements for
different ecosystem types to calibrate and evaluate the models. CO flux measurements
are mostly from short-term field observations or laboratory experiments (e.g. Conrad
and Seiler, 1985a; Funk et al., 1994; Tarr et al., 1995; Zepp et al., 1997; Kuhlbusch et
al., 1998; Moxley and Smith, 1998; Schade et al., 1999; King and Crosby, 2002; Varella
et al., 2004; Lee et al., 2012; Bruhn et al., 2013; van Asperen et al., 2015). The first
study to report long-term and continuous field measurements of CO flux over grassland
using a micrometeorological eddy covariance (EC) method are in Pihlatie et al. (2016).
Aiming to improve the understanding of processes associated with land-
atmosphere CO exchange and to quantify global soil CO budget for the 20th and 21st
centuries, we developed a CO dynamics module (CODM) embedded in a process-
based biogeochemistry model, the Terrestrial Ecosystem Model (TEM) (Zhuang et al.,
2003, 2004, 2007). CODM was then calibrated and evaluated using laboratory
experiments and field measurements for different ecosystem types. We then used the
atmospheric CO concentration data from MOPITT (Gille, 2013) to drive our model from
2000 to 2013.   We conducted century-long simulations of 1901-2100, using the
atmospheric CO concentrations estimated withan empirical function (Badr & Probert,
1994; Potter et al., 1996). We also evaluated the effects of multiple forcings on global
CO consumption and production estimates, including the changes of climate and
atmospheric CO concentrations at the global scale.





## 2. Method

### 2.1 Overview

We first developed a daily soil CO dynamics module (CODM) that considers: (1)
soil-atmosphere CO exchange and diffusion process between soil layers, (2)
consumption by soil microbial oxidation, (3) production by soil chemical oxidation, and
(4) the effects of temperature, soil moisture, soil CO substrate and surface atmospheric
CO concentration on these processes. Second, we used observed soil temperature and
moisture to evaluate TEM hydrology module and soil thermal module in order to
estimate soil physical variables correctly. Then we used results of laboratory
experiments and CO flux measurements to parameterize the model and calibrate the
model using the Shuffled Complex Evolution (SCE-UA) method (Duan et al., 1993).
Finally, the model was extrapolated to the global scale at a 0.5° by 0.5° resolution. We
conducted three sets of model experiments to investigate the impact of climate and
atmospheric CO concentrations on soil CO dynamics: 1) 1901-2013 with constant
atmospheric CO concentrations estimated from an empirical function; 2) 2000-2013 with
MOPITT satellite atmospheric CO concentration data; and 3) 2014-2100 with the same
constant atmospheric CO concentrations as 1) and three future climate scenarios.

### 2.2 Carbon Monoxide Dynamics Module (CODM)

Embedded in TEM (Figure 1), CODM is mainly driven by: (1) soil organic carbon
availability based on a carbon and nitrogen dynamics module (CNDM) (Zhuang et al.,
2003); (2) soil temperature profile from a soil thermal module (STM) (Zhuang et al.,
2001, 2003); and (3) soil moisture profile from a hydrological module (HM) (Bonan,
1996; Zhuang et al, 2004). Net exchange of CO between the atmosphere and soil is
determined by the mass balance. According to previous studies, we separate active
soils (top 30cm) for CO consumption and production into 1 cm thick layers (King, 1999a,
1999b; Whalen & Reeburgh, 2001; Chan & Steudler, 2006). Between the soil layers, the
changes of CO concentrations are calculated by:

$$\frac{\partial (C(t,i))}{\partial t} = \frac{\partial}{\partial z}\left(D(t,i)\frac{\partial\big(C(t,i)\big)}{\partial z}\right) + P(t,i) - O(t,i) \qquad (1)$$





Where $C(t, i)$ is the CO concentration in layer i and time t, units are mg m$^{-3}$. $z$ is the
thickness of layer $i$. $D(t, i)$ is the diffusion coefficient for layer $i$, units are m$^2$ s$^{-1}$. $P(t, i)$
is the CO production rate and $O(t, i)$ is CO consumption rate due to oxidation. The units
of $P(t, i)$ and $O(t, i)$ are mg m$^{-3}$ s$^{-1}$. $D(t, i)$ is calculated using the method from Potter et
al. (1996), equation (2) to (4), which is the function of soil temperature, soil texture and
soil moisture. The upper boundary condition is specified as the atmospheric CO
concentration, which is estimated by an empirical function of latitude (Potter et al., 1996)
or directly measured by the MOPITT satellite during 2000-2013. The lower boundary
condition is assumed to have no diffusion exchange with the layer underneath. This
partial differential equation (PDE) is solved using the Crank-Nicolson method for less
time-step-sensitive solution.
CO consumption is modeled as an aerobic process occurring in unsaturated soil
pores, which is estimated as:

$$O(t, i) = V_{max} \cdot f_1\big(C(t, i)\big) \cdot f_2\big(T(t, i)\big) \cdot f_3\big(M(t, i)\big) \quad (2)$$

Where $V_{max}$ is the specific maximum oxidation rate, ranging from 0.3 to 11.1 µg CO g$^{-1}$
h$^{-1}$ (Whalen & Reeburgh, 2001). $f_i$ are functions calculating CO concentration $C(t, i)$,
temperature $T(t, i)$ and moisture $M(t, i)$ influences on CO soil consumption.
Considering CO consumption as the result of microbial activities, we calculate
$f_1\big(C(t, i)\big)$, $f_2\big(T(t, i)\big)$ and $f_3\big(M(t, i)\big)$ in a similar way as Zhuang et al. (2004):

$$f_1\big(C(t, i)\big) = \frac{C(t, i)}{C(t, i) + k_{CO}} \quad (2.1)$$

$$f_2\big(T(t, i)\big) = Q_{10}^{\frac{T(t,i) - T_{ref}}{10}} \quad (2.2)$$

$$f_3\big(M(t, i)\big) = \frac{(M(t, i) - M_{min})(M(t, i) - M_{max})}{(M(t, i) - M_{min})(M(t, i) - M_{max}) - (M(t, i) - M_{opt})^2} \quad (2.3)$$

Where $f_1\big(C(t, i)\big)$ is a multiplier that enhances oxidation rate with increasing soil CO
concentrations using a Michaelis-Menten function with a half-saturation constant $k_{CO}$,
ranging from 5 to 51 µl CO l$^{-1}$ (Whalen & Reeburgh, 2001); $f_2\big(T(t, i)\big)$ is a multiplier that
enhances CO oxidation rates with increasing soil temperature using a Q10 function with
$Q_{10}$ coefficients (Whalen & Reeburgh, 2001). $T_{ref}$ is the reference temperature, units





are °C (Zhuang et al., 2004, 2013). $f_3\big(M(t,i)\big)$ is a multiplier to estimate the biological
limiting effect that diminishes CO oxidation rates if the soil moisture is not at an optimum
level ($M_{opt}$). $M_{min}$, $M_{max}$ and $M_{opt}$ are the minimum, maximum and optimum volumetric
soil moistures of oxidation reaction, respectively. Equation (2.2) will overestimate CO
consumption at higher temperature because CO consumption has an optimum
temperature and it will decrease at higher temperatures. However, the CO consumption
is constrained by CO production, and equation (1) is used to represent this constraint.

We model the CO production rate ($P(t,i)$) as a process of chemical oxidation

constrained by soil organic carbon (SOC) decay (Conrad and Seiler,1985; Potter et al.
1996; Jobbagy & Jackson, 2000; van Asperen et al., 2015):

$$P(t,i) = P_r(t,i) \cdot E_{SOC} \cdot C_{SOC}(t) \cdot F_{SOC} \qquad (3)$$

Where $P_r(t,i)$ is a reference soil CO production rate which has been normalized to rate
at reference temperature, which is affected by soil moisture and soil temperature
(Conrad and Seiler,1985; van Asperen et al., 2015). $E_{SOC}$ is an estimated nominal CO
production factor of $3.5 \pm 0.9 \times 10^{-9}$ mg CO m$^{-2}$ s$^{-1}$ per g SOC m$^{-2}$ (to 30 cm surface soil
depth) (Potter et al., 1996). $C_{SOC}(t)$ is a SOC content in mg m$^{-2}$, which is provided by
CNDM module in TEM. $F_{SOC}$ is a constant fraction of top 20cm SOC compared to total
amount of SOC, which is 0.33 for shrubland areas, 0.42 for grassland areas and 0.50
for forest areas, respectively (Jobbagy & Jackson, 2000). $P_r(t,i)$ is calculated as:

$$P_r(t,i) = \exp\left( f_4\big(M(t,i)\big) \cdot Ea_{ref}/R \cdot \left( \frac{1}{273.15 + PT_{ref}} - \frac{1}{T(t,i) + 273.15} \right) \right) \qquad (3.1)$$

$$f_4\big(M(t,i)\big) = \frac{PM_{ref}}{M(t,i) + PM_{ref}} \qquad (3.2)$$

Where equation (3.1) is derived from Arrhenius equation for chemical reactions and
normalized using the reference temperature $PT_{ref}$. $Ea_{ref}/R$ is the reference activation
energy divided by gas constant $R$, units are K. $f_4\big(M(t,i)\big)$ is the multiplier that reduces
activation energy using an regression approach based on laboratory experiment of
moisture influences on CO production (Conrad and Seiler,1985). $PM_{ref}$ is the reference
volumetric soil moisture, ranging from 0.01 to 0.5 volume/volume (v/v).  We assume





thermal-degradation as the main CO producing process since lack of photo-degradation
data and hard to distinguish photo-degradation from observations. In order to reduce
the bias from thermal-degradation to total abiotic degradation, the equation (3.1) is
parameterized by comparing with total production rate. For instance, $P_r(t, i)$ calculation
can perfectly fit the experiment results in Van Asperen et al., 2015 with proper
$PT_{ref}$(18°C), $Ea_{ref}/R$(14000 K)  and $PM_{ref}$(0.5 v/v).

**2.3 Model Parameterization and Extrapolation**

The model parameterization was conducted in two steps: 1) Thermal and

hydrology modules embedded in TEM were revised, calibrated and evaluated by
running model with corresponding local meteorological or climatic data at 4
representative sites, including boreal forest, temperate forest, tropical forest and
savanna (Table 1, site No.1 to 4, Figure 2) to minimize model data misfit in terms of soil
temperature and moisture. 2) CODM module was parameterized by running TEM for
observational periods with the corresponding local meteorological or climatic data at
each reference site (Table 1, Figure 3), and using Shuffled Complex Evolution
Approach in R language (SCE-UA-R) (Duan et al., 1993) to minimize the difference
between simulated and observed net CO flux. Eleven parameters including $k_{CO}$, $V_{max}$,
$T_{ref}$, $Q_{10}$, $M_{min}$, $M_{max}$, $M_{opt}$, $E_{SOC}$, $Ea_{ref}/R$, $PM_{ref}$ and $PT_{ref}$ are optimized (Table 2). To
be noticed, $F_{SOC}$ was not involved in the calibration process. Parameter priors were
decided based on previous studies (Conrad & Seiler, 1985; King, 1999b; Whalen &
Reeburgh, 2001; Zhuang et al., 2004).  SCE-UA-R was used for site No. 6, 8, 10, 11
(Table 1). Each site had been run 50 times using SCE-UA-R with 10000 maximum
loops for parameter ensemble, and all of them reached stable state before the end of
the loops.  For wetlands, the only available data is from site No.12. We used trial-and-
error method instead to make our simulated results in the range of observed flux rates,
with a 10% tolerance. For tropical sites, since tropical savanna vegetation type is a
combination type of tropical forest and grassland in our model, we first used Site No. 13
to set priors to fit the experiment results with a 10% tolerance and then evaluated by
running our model comparing with site No.7 results. Site No. 9 and 5 were used to
evaluate our model results for temperate forest and grassland. Besides the observed





climatic and soil property data, we used ERA-Interim reanalysis data from The
European Centre for Medium-Range Weather Forecasts (ECMWF) (Dee et al., 2011),
AmeriFlux observed meteorology data (http://ameriflux.lbl.gov/) and reanalysis climatic
data from Climatic Research Unit (CRU, Harris et al., 2013) to fill the missing
environmental data. To sum up, parameters for various ecosystem types in table 2 were
the final results of our parameterization. Model parameterization was conducted for
ecosystem types including boreal forest, temperate coniferous forest, temperate
deciduous forest, and grassland using SCE-UA-R. Tropical forest and wet tundra used
a trial-and-error method to adjust parameters letting simulation results best fit the lab
data. Due to limited data availability, we assumed temperate evergreen broadleaf forest
having the same parameters as temperate deciduous forest.

**2.4 Data Organization**
To get spatially and temporally explicit estimates of CO consumption, production
and net flux at the global scale, we used the data of land cover, soils, climate and leaf
area index (LAI) from various sources at a spatial resolution of 0.5° latitude X 0.5°
longitude to drive TEM. The land cover data include potential vegetation distribution
(Melillo et al., 1993) and soil texture (Zhuang et al., 2003), which were used to assign
vegetation- and texture-specific parameters to each grid cell.
For the simulation of the period 1901-2013, monthly air temperature, precipitation,
clouds fraction and vapor pressure data sets from CRU were used to estimate the soil
temperature, soil moisture and SOC with TEM (Figure 4). Monthly LAI data from TEM
were required to simulate soil moisture (Zhuang et al., 2004). During this period time,
we used an empirical function of latitude, which was derived from the observed
latitudinal distribution of tropospheric carbon monoxide (Badr and Probert, 1994) to
calculate CO surface concentrations (equation (7), Potter et al., 1996):

$$C_{CO,air} = 82.267856 + 0.8441503L + 1.55934 \times 10^{-2}L^2 + 2.37 \times 10^{-5}L^3 - 2.3 \times 10^{-6}L^4$$

Where $C_{CO,air}$ is the derived surface CO concentration (ppbv), *L* represents
latitude which is negative degrees for southern hemisphere and positive degrees for
northern hemisphere. We also used the transient atmospheric CO data from MOPITT





satellite during 2000-2013 (Figure 5). We averaged day-time and night-time monthly
mean retrieved CO surface level 3 data (variables mapped on 0.5° latitude X 0.5°
longitude grid scales with monthly time step, Gille, 2013) to represent the CO surface
concentration level in each month. The missing pixels were fixed by the average of
pixels which had values and were inside 1.5 times of the distance between this missing
pixel and the nearest pixel with values. These global mean values shown in Figure 5 do
not include ocean surfaces, thus there are differences between our surface CO
concentration results and Yoon and Pozzer's report in 2014, which is as low as 99.8ppb.
From 2014 to 2100, we used Intergovernmental Panel on Climate Change (IPCC) future
climate scenarios from Representative Concentration Pathways (RCPs) climate forcing
data sets RCP2.6, RCP4.5 and RCP8.5 (Figure 6). Since RCPs did not have water
vapor pressure data, we use the specific humidity and sea level air pressure from the
RCPs and elevation of surface to estimate the monthly surface vapor pressure data
(Seinfeld & Pandis, 2006).

**2.5 Model Experiment Design**

We conducted two core simulations and eight sensitivity test simulations in

historical period. The two core simulations were driven with CO surface concentrations
estimated from an empirical function of latitude (experiment E1) for the period 1901-
2013 and with transient CO surface concentrations from MOPITT satellite data
(experiment E2) for the period 2000-2013, respectively. Eight sensitivity simulations
were driven with constant CO surface concentrations ± 30%, SOC ±30%, precipitation
±20% and air temperature ± 3°C for each pixel during 1999-2000 (E3). For the 21st
century, we conducted simulations driven with climate data of RCP2.6, RCP4.5 and
RCP8.5 to examine the responses of CO flux to changing climates (E4).

**3.  Results**
**3.1 Site Evaluation**

Both the magnitude and variation of the simulated soil temperature and moisture

from cold area to warm area compared well to the observations (Figure. 2). The





magnitude of simulated CO flux is highly correlated with the observations (r is about 0.5,
p-value < 0.001, Figure 3, a2, b2 ,c2 ,d2). Root mean square error (RMSE) of the
simulated CO flux for all sites is below 1.5 mg CO m$^{-2}$ day$^{-1}$. RMSE for site No. 7 is
bigger than 2.0 mg CO m$^{-2}$ day$^{-1}$ when compared with transparent chamber
observations. For boreal forest site, we only have 8 acceptable points in 1994 and 1996
(Figure 3c2).
**3.2 Global Soil CO Dynamics During 1901-2013**
For the simulation with constant CO surface concentrations (E1) during 1901-
2013, the estimated mean soil CO consumption, production and net flux (positive
direction is from soil to atmosphere) are -141, 32 and -108 Tg CO yr$^{-1}$, respectively. In
the long-term simulations, annual soil CO fluxes vary slightly. The annual soil CO
consumption, production and net flux vary within 10% during the period (Figure 8a).
Consumption is about 4 times larger than production. The highest rates of consumption
and production are located in areas close to the equator, and consumption from areas
such as eastern US, Europe and eastern Asia also has large rates (>-1000 mg m$^{-2}$ yr$^{-1}$)
(Figure 7a, b). Globally soils serve as atmospheric CO sink (Figure 7c). Some areas,
such as western US and southern Australia, are CO sources, all of which are grassland
or experiencing dry climate. The latitudinal distributions of consumption, production and
net flux rates share the same spatial pattern. Around 20°S-20°N and 20-60N° are the
largest and second largest areas for production and consumption, while the 45°S-45°N
area accounts for nearly 90% of total consumption and production (Figure 9a, Table 3).
The Southern and Northern Hemispheres consume 42% and 58% of the total
consumption, and produce 41% and 59% of total production, respectively (Table 3).
Tropical evergreen forests are the largest sinks, consuming 66 Tg CO yr$^{-1}$, and tropical
savanna and deciduous forest are second and third largest sinks, consuming a total of
27 Tg CO yr$^{-1}$ (Table 4). These three ecosystems account for 66% of the total
consumption. Tropical evergreen forests are also the largest source of soil CO
production, producing 15 Tg CO yr$^{-1}$, while tropical savanna have a considerable
production 6 Tg CO yr$^{-1}$ (Table 4). Moreover, tropical areas, including forested wetlands,
forested floodplain and evergreen forests, are most efficient for CO consumption,



ranging from -10 to -12 mg CO m$^{-2}$ day$^{-1.}$ They are also the most efficient for CO
production at over 2 mg CO m$^{-2}$ day$^{-1}$ (Table 3).
For the simulation with transient atmospheric CO surface concentrations (E2)
during 2000-2013, the mean annual global soil consumption increases to -187 Tg CO
yr$^{-1}$, and areas near the equator become large sinks for atmospheric CO together with
eastern US, Europe, and eastern Asia (Figure 7) due to the heavy atmospheric CO
burden over these areas (Figure 5a). The annual consumption and net flux trends follow
the atmospheric CO concentration trends (Figure 5b, Figure 8b), with a small
interannual variability (<10%). The latitudinal distributions of soil CO fluxes for E1 and
E2 are similar but E2's CO fluxes magnitudes are larger than E1's and around 30°N of
E2's distribution shows another peak of CO consumption, due to the high atmospheric
CO concentration over eastern Asia (Figure 5a, Figure 9b). The consumption between
45°S-45°N increases by 35%, to -137 Tg CO yr$^{-1}$, which is 73% of the global total
annual consumption. Consumption rates of high latitude areas (45°N North) do not
change significantly (Figure 7, 9, Table 3), and the annual consumption only increases
by 10%, thus the portion of soil CO sinks in northern high latitudes decreases from 12%
to 10% of the global total.

**3.3 Global Soil CO Dynamics During 2014-2100**
Using the constant atmospheric CO, the estimated annual mean soil CO
consumptions for the period 2014-2100 are -162, -174 and -194 Tg CO yr$^{-1}$ while
estimated annual mean soil productions are 36, 39 and 44 Tg CO yr$^{-1}$ for RCP2.6, 4.5
and 8.5 scenarios, respectively.  The net fluxes are -118.06, -117.31 and -115.13 Tg
CO yr$^{-1}$ at the beginning 10 years of the 21$^{st}$ century, and will reach -127.17, -144.99
and -187.25 Tg CO yr$^{-1}$ at the end of the 21$^{st}$ century for RCP2.6, RCP4.5 and RCP8.5
scenarios, respectively (Figure 11). Global distribution patterns of CO consumption,
production and net flux are similar to the 20th century but there are significant
differences among RCP2.6 RCP4.5 and 8.5 scenarios on areas near the equator, flux
rates increasing from RCP2.6 to 8.5. Areas near the equator and eastern Asia become
big sinks of atmospheric CO, while northeastern US becomes a small source (Figure
10). The consumption has relatively fast growth rates during the 21$^{st}$ century (Figure 11).



Furthermore, there are significant trends of increasing consumption, production and net
flux for nearly all scenarios. The rate ranges of increasing of consumption, production,
and net flux are -0.15 to -1.23, 0.04 to 0.3, and -0.12 to 0.94 Tg CO $yr^{-2}$, respectively
(Figure 11). These increasing trends are similar to air temperature increasing trends
(Figure 6).

**4. Discussion**
**4.1 Comparison with Other Studies**
Previous studies estimated a large range of global CO consumption from -16 to -
636 Tg CO $yr^{-1}$. Our estimates are -132 to -154 Tg CO $yr^{-1}$ for the 20th century and -180
to -197 Tg CO $yr^{-1}$ for 2000-2013 using MOPITT satellite CO surface concentration data.
Previous studies also provide a large range for CO production from 0 to 7.6 mg $m^{-2}$ $day^{-1}$
(reviewed in Pihlatie et al., 2016). Our results showed averaged CO production
ranging from 0.01 to 2.29 mg $m^{-2}$ $day^{-1}$. The large uncertainty of these estimates is
mainly due to a different consideration of the microbial activities, the depth of the soil,
and the parameters in the model. In contrast to the estimates of -16 to -57 Tg CO $yr^{-1}$
which were based on top 5 cm soils (Potter et al., 1996), our estimates considered
30cm soils, just as used in Whalen & Reeburgh (2001). In addition, we used a thinner
layer division (1cm each layer) for diffusion process, and used the Crank-Nicolson
method solving partial differential equations to avoid time step influences. We also
included microbial CO oxidation process to remove the CO from soils and the effects of
soil moisture, soil temperature, vegetation type and soil CO substrate on microbial
activities. Besides, our soil thermal, soil hydrology and carbon and nitrogen dynamics
simulated in TEM provide carbon substrate spatially and temporally for estimating soil
CO dynamics (Bonan, 1996; Zhuang et al., 2001, 2003, 2004, 2007). Overall, although
a few previous studies have examined the long-term impacts of climate, land use and
nitrogen depositions on CO dynamics (Chan & Steudler, 2006, Pihlatie et al., 2016 ),
global prediction of soil CO dynamics still have a large uncertainty.

**4.2 Major Controls to Soil CO Dynamics**





Eight sensitivity tests have been conducted for the 1999-2000 period, including
changing atmospheric CO by ±30%, SOC by ±30%, precipitation by ±30% and air
temperature by ±3°C for each pixel (Table 5). Soil CO consumption is most sensitive
(changing 29%) to air temperature while production is most sensitive (changing up to
36%) to both air temperature and SOC (30%). The net CO fluxes have the similar
sensitivities to consumption, because consumption is normally much larger than CO
production so that it will determine the dynamics of the net flux. Annual CO consumption,
production and net flux follow the change of air temperature (Table 5), which explains
the small increasing trends after the 1960s, the significant increasing trend in the 21$^{st}$
century and the large sinks over tropical areas. Besides, a 30% change in precipitation
will not lead to large changes in CO flux (< 3%). SOC did not directly influence CO
consumption. Increasing SOC led to an increase in soil CO substrate so implying that
more CO in soils can be consumed. CO concentrations will only influence the uptake
rate and soil CO substrate concentrations, thus influencing the soil CO consumption
rate.
Annual CO consumption, production and net flux are significantly correlated with
air temperature and soil temperature, due to increasing microbial activities (R > 0.91
globally). Specifically, annual CO production is strongly correlated with annual mean
SOC. The annual mean SOC follows air temperature trends (Figure 4) as CO flux.
Consumption has low correlations with annual precipitation and soil moisture, especially
at 45°N-45°S (Table 6). The soil moisture is significantly influenced by temperature
since increasing temperature would result in higher evapotranspiration. In contrast, the
monthly consumption and production are correlated with the precipitation and soil
moisture in the Northern Hemisphere (R>0.85), which contains over 53% of the global
soil CO consumption (Table 3). Meanwhile, the monthly CO flux is still well correlated
with air temperature and soil moisture. Monthly CO flux has low correlations with  SOC
because the soil organic carbon will not change greatly within a month. The correlation
between annual soil CO consumption and atmospheric CO concentration is 0.91 at the
global scale because the atmospheric CO concentration, air temperature, soil
temperature dominate the annual consumption rate. At monthly step, this correlation is -
0.48 because global atmospheric CO concentrations are high in winter and low in



summer while the simulated soil CO consumption shows an opposite monthly variation(Table 6, Figure 12), suggesting that other factors such as precipitation, air temperature, and soil temperature are major controls for monthly CO fluxes.

**4.3. Model Uncertainties & Limitations**

Due to the lack of long-period observational data of CO flux and associated environmental factors, the model parameterization using SCE-UA-R method can only be done for 4 ecosystem types including boreal forest, temperate coniferous forest, temperate deciduous forest and grassland, with RMSE ranging from 0.56 to 1.47 mg m$^{-2}$ day$^{-1}$. Tropical forest calibration is only conducted using a very limited amount of lab experiment data, but tropical areas are hotspots for CO soil-atmosphere exchange. Besides, tropical forest SOC for top 30cm can be really high according to observations. TEM model may underestimate the top 30cm SOC, which will underestimate production rates, especially in tropical region. The large deviation for tropical savanna (which is mosaic of tropical forest and grassland ecosystems) may be due to using outside air temperature to represent inside air temperature of transparent chamber observations (Varella et al., 2004), and uncertain tropical forest parameterization. We used the conclusion from van Asperen et al. (2015) and only considered the thermal-degradation process for CO production. Photo-degradation process and biological formation process were not considered due to lacking understanding of these processes. Although we focused on natural ecosystems in this study, land-use change, agriculture activity, and nitrogen deposition also affect the soil CO consumption and production (King, 2002; Chan & Steudler, 2006). For instance, soil CO consumption in agriculture ecosystems is 0 to 9 mg CO m$^{-2}$ day$^{-1}$ in Brazil (King & Hungria, 2002). We used grass land or forest ecosystem to represent agriculture areas in CODM module. Our future study shall include these processes and factors.

**5. Conclusions**

We analyzed the magnitude, spatial pattern, and the controlling factors of atmosphere-soil CO exchange at the global scale for the 20th and 21st centuries using





a calibrated process-based biogeochemistry model. Major processes include
atmospheric CO diffusion into soils, microbial oxidation removal of CO, and CO
production through chemical reaction. We found that air temperature and soil
temperature play a dominant role in determining annual soil CO consumption and
production while precipitation, air temperature, and soil temperature are the major
controls for the monthly consumption and production. Atmospheric CO concentrations
will be important for annual CO consumption. We estimated that the global annual CO
consumption, production and net flux for the 20th century are 132-154, 29-36 and 112-
119 Tg CO yr$^{-1}$, respectively, when using a constant atmospheric CO concentration. The
CO consumption reaches 180-197 Tg CO yr$^{-1}$ during 2000-2013 when using
atmospheric CO concentrations observed by the MOPITT satellite. Tropical evergreen
forest, savanna and deciduous forest areas are the largest sinks accounting for 66% of
the total CO consumption, while the Northern Hemisphere consumes 60% of the global
total. During the 21st century, the predicted net CO flux will reach 126-150 Tg CO yr$^{-1}$ in
the 2090s, primarily because of increasing air temperature. The areas near the equator,
eastern Asia, Europe and eastern US will become the sink hotspots because they have
warm and moist soils. This study calls for long-period observations of CO flux for
various ecosystem types to improve models. The effects of land-use change, agriculture
activities, nitrogen deposition, photo-degradation and biological formation shall also be
considered to improve future quantification of soil CO fluxes.


**Acknowledgment**

This study is supported through projects funded to Q.Z. by Department of Energy

(DE-SC0008092 and DE-SC0007007) and the NSF Division of Information and
Intelligent Systems (NSF-1028291). The supercomputing resource is provided by
Rosen Center for Advanced Computing at Purdue University. We acknowledge Dr.
Stephen C. Whalen made the observational CO flux data available to this study. We are
also grateful to University of Tuscia (dep. DIBAF), Italy, and their affiliated members, for
their help and the use of their field data.



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





**Table 1.** Model parameterization sites for thermal and hydrology modules (site No. 1-4) and for CODM module (site No. 5-13)

| No. | Site Name | Location | Vegetation | Driving Climate | Observed Data | Source and Comments |
|---|---|---|---|---|---|---|
| 1 | Poker Flat Research Range Black Spruce Forest (US_PRR) | 147°29'W/65°7'N | Boreal Evergreen Needle Leaf Forests | Site Observation & ERA Interim | Soil Temperature and Moisutre of 2011-2014 | Suzuki (2016) |
| 2 | Morgan Monroe State Forest (US_MMS) | 86°25'W/39°19'N | Temperate Deciduous Broadleaf Forests | Site Observation & ERA Interim | Soil Temperature and Moisutre of 1999-2014 | Philip and Novick (2016) |
| 3 | Santarem, Tapajos National Forest (STM_K83) | 54°56'W/3°3'S | Tropical Moist Forest | Site Observation & ERA Interim | Soil Temperature and Moisutre of 2000-2004 | SALESKA et al. (2013) |
| 4 | Bananal Island Site (TOC_BAN) | 50°08'W/9°49'S | Tropical Forest-Savanna | Site Observation & ERA Interim | Soil Temperature and Moisutre of 2003-2006 | SALESKA et al. (2013) |
| 5 | Eastern Finland (EF) | 27°14'E/63°9'N | Boreal Grassland | Site Observation & ERA Interim | CO flux of April-November,2011 | Pihlatie et.al. (2016) |
| 6 | Viterbo, Italy (VI) | 11°55'E/42°22'N | Mediterranean Grassland | Site Observation & ERA Interim | CO flux of August, 2013 | van Asperen et al. (2015) |
| 7 | Brasilia, Brazil (BB) | 47°51'W/15°56'S | Tropical Savanna | Site Observation & CRU | CO flux of October 1999 to July 2001 | Varella et al. (2004) |
| 8 | Orange County, North Carolina (OC) | 79°7'W/35°58'N | Temperate Coniferous Forest | AMF_US-Dk3 2002-2003 | CO flux of March 2002 to March 2003 | Fisher (2003) |
| 9 | Tsukuba Science City, Japan (TSC) | 140°7'E/36°01'N | Temperate Mixed Forest | Site Observation & ERA Interim | CO flux of July 1996 to September 1997 | Yonemura et. al. (2000) |
| 10 | Manitoba, Canada (CBS) | 96°44'W/56°09'N | Boreal Pine Forest | Site Observation & AMF_CA-Man | CO flux of June-August, 1994 | Kuhlbusch et. al (1998) |
| 11 | Scotland, U.K. (SUK) | 3°12'W/55°51'N | Temperate Deciduous Forests | ERA Interim 1995 | CO flux of 1995 | Moxley and Smith (1998) |
| 12 | Alaska, USA (AUS) | 147°41'W/64°52'N | Boreal wetland | CRU 1991 | CO flux of Lab Experiment,1991 | Funk et al. (1994) |
| 13 | Guayana Shield,Bolivar State,Venezuela (GBV) | 62°57'W/7°51'N | Tropical Smideciduous Forest | CRU 1985 | CO flux of Lab Experiment,1985 | Scharffe et al. (1990) |

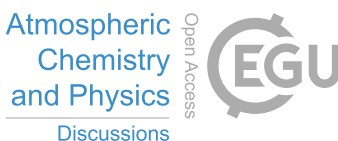



**Table 2.** Ecosystem-specific parameters in the CODM module[a]

| | Ecosystem Type | $k_{CO}$ (ul CO $l^{-1}$) | $V_{max}$ ($\mu g$ CO $g^{-1} h^{-1}$) | $T_{ref}$ (°C) | $Q10$ (Unitless) | $M_{min}$ ($\frac{v}{v}$) | $M_{max}$ ($\frac{v}{v}$) | $M_{opt}$ ($\frac{v}{v}$) | $E_{soc}$ | $F_{soc}$ ($\frac{g}{g}$) | $\frac{Ea_{ref}}{R}$ (K) | $PM_{ref}$ ($\frac{v}{v}$) | $PT_{ref}$ (°C) |
|---|---|---|---|---|---|---|---|---|---|---|---|---|---|
| 1 | Alpine Tundra & Polar Desert | 36.00 | 0.78 | 4.00 | 1.80 | 0.10 | 1.00 | 0.55 | 3.00 | 0.33 | 7700 | 0.25 | 30.00 |
| 2 | Wet Tundra | 36.00 | 0.70 | 4.00 | 1.80 | 0.25 | 1.00 | 0.55 | 3.00 | 0.42 | 7700 | 0.25 | 30.00 |
| 3 | Boreal Forest | 27.34 | 1.18 | 9.81 | 1.60 | 0.15 | 0.64 | 0.53 | 2.98 | 0.50 | 8827 | 0.35 | 26.99 |
| 4 | Temperate Coniferous Forest | 42.64 | 2.15 | 6.90 | 1.87 | 0.02 | 0.96 | 0.53 | 2.86 | 0.50 | 8404 | 0.38 | 31.52 |
| 5 | Temperate Deciduous Forest | 40.16 | 2.43 | 8.54 | 1.51 | 0.17 | 0.81 | 0.51 | 2.45 | 0.50 | 8801 | 0.35 | 37.44 |
| 6 | Grassland | 42.41 | 0.49 | 11.27 | 1.65 | 0.16 | 0.82 | 0.51 | 3.09 | 0.42 | 14165 | 0.24 | 12.29 |
| 7 | Xeric Shrublands | 8.00 | 0.30 | 4.00 | 1.50 | 0.10 | 1.00 | 0.55 | 3.00 | 0.33 | 7700 | 0.25 | 30.00 |
| 8 | Tropical Forest | 45.00 | 2.00 | 4.00 | 1.50 | 0.10 | 1.00 | 0.55 | 3.80 | 0.50 | 14000 | 0.50 | 18.00 |
| 9 | Xeric Woodland | 8.00 | 0.30 | 4.00 | 1.50 | 0.10 | 1.00 | 0.55 | 3.00 | 0.50 | 7700 | 0.25 | 30.00 |
| 10 | Temperate Evergreen Broadleaf Forest | 40.16 | 2.43 | 8.54 | 1.51 | 0.17 | 0.81 | 0.51 | 2.45 | 0.50 | 8801 | 0.35 | 37.44 |
| 11 | Mediterranean Shrubland | 45.00 | 1.50 | 4.00 | 1.50 | 0.10 | 1.00 | 0.55 | 3.00 | 0.33 | 7700 | 0.25 | 30.00 |
| ** | Largest Potential Value | 51.00 | 11.1 | 15.00 | 2.00 | 0.30 | 1.00 | 0.60 | 3.80 | -- | 15000 | 0.60 | 40.00 |

[a] $k_{CO}$ is the half-saturation constant for soil CO concentration; $V_{max}$ is the specific maximum CO oxidation rate; $T_{ref}$ is the reference temperature to account for soil temperature effects on CO consumption; $Q10$ is the an ecosystem-specific Q10 coefficient to account for soil temperature effects on CO consumption; $M_{min}$, $M_{max}$, $M_{opt}$ are the minimum, optimum, and maximum volumetric soil moistures of oxidation reaction to account for soil moisture effects on CO consumption; $E_{soc}$ is an estimated nominal CO production factor, similar as Potter et al. (1996) ($10^{-4}$ mg CO $m^{-2}$ $d^{-1}$ per g SOC $m^{-2}$); $F_{soc}$ is a constant fraction of top 20cm SOC compared to total amount of SOC to account for SOC effects on CO production; $Ea_{ref}/R$ is the is the ecosystem-specific activation energy divided by gas constant to account for the reaction rate of production; $PM_{ref}$ is the reference moisture to account for soil temperature effects on CO production; $PT_{ref}$ is the reference temperature to account for soil temperature effects on CO production






**Table 3.** Regional soil CO consumption, net flux and production(Tg CO yr$^{-1}$) during 1901- 2013 (E1) and during
2000-2013 withMOPITT data transient CO surface concentration

| | South-45°S | | 45°S-0° | | 0°-45°N | | 45°N-North | | Global | |
|---|---|---|---|---|---|---|---|---|---|---|
| | E1[a] | E2[b] | E1 | E2 | E1 | E2 | E1 | E2 | E1 | E2 |
| Consumption | -0.20 | -0.22 | -58.36 | -75.77 | -64.78 | -91.66 | -17.24 | -18.90 | -140.58 | -186.55 |
| Net flux | -0.12 | -0.13 | -43.39 | -59.34 | -51.58 | -77.17 | -13.27 | -14.63 | -108.35 | -151.27 |
| Production | 0.09 | 0.09 | 14.98 | 16.43 | 13.20 | 14.49 | 3.97 | 4.27 | 32.23 | 35.28 |


[a]E1 represents the simulation with constant CO surface concentration data;
[b]E2 represents the simulation with MOPITT transient CO surface concentration data.
























**Table 4.** Global soil CO consumption, net flux and production in different ecosystems during 1901-2013

| Vegetation Type | Area (10^6 km²) | Pixels | Consumption (Tg CO yr⁻¹) | Net flux (Tg CO yr⁻¹) | Production (Tg CO yr⁻¹) |
|---|---|---|---|---|---|
| Alpine Tundra & Polar Desert | 5.28 | 3580 | -0.95 | -0.73 | 0.22 |
| Wet Tundra | 5.24 | 4212 | -1.06 | -0.52 | 0.54 |
| Boreal Forest | 12.47 | 7578 | -7.19 | -5.53 | 1.66 |
| Forested Boreal Wetland | 0.23 | 130 | -0.13 | -0.09 | 0.04 |
| Boreal Woodland | 6.48 | 4545 | -2.40 | -1.53 | 0.88 |
| Non-Forested Boreal Wetland | 0.83 | 623 | -0.33 | -0.17 | 0.16 |
| Mixed Temperate Forest | 5.25 | 2320 | -6.31 | -5.82 | 0.49 |
| Temperate Coniferous Forest | 2.49 | 1127 | -2.78 | -2.49 | 0.29 |
| Temperate Deciduous Forests | 3.65 | 1666 | -3.26 | -3.02 | 0.24 |
| Temperate Forested Wetland | 0.15 | 60 | -0.22 | -0.21 | 0.01 |
| Tall Grassland | 3.63 | 1567 | -1.28 | -0.37 | 0.91 |
| Short Grassland | 4.71 | 2072 | -0.93 | -0.25 | 0.68 |
| Tropical Savanna | 13.85 | 4666 | -15.83 | -10.41 | 5.42 |
| Xeric Shrubland | 14.71 | 5784 | -1.61 | -1.33 | 0.28 |
| Tropical Evergreen Forest | 17.77 | 5855 | -66.12 | -51.28 | 14.84 |
| Tropical Forested Wetland | 0.55 | 178 | -2.51 | -2.05 | 0.46 |
| Tropical Deciduous Forest | 4.69 | 1606 | -11.20 | -8.48 | 2.72 |
| Xeric Woodland | 6.85 | 2387 | -6.53 | -5.57 | 0.96 |
| Tropical Forested Floodplain | 0.15 | 50 | -0.64 | -0.54 | 0.11 |
| Desert | 11.61 | 4170 | -0.49 | -0.45 | 0.05 |
| Tropical Non-forested Wetland | 0.06 | 19 | -0.02 | -0.01 | 0.01 |
| Tropical Non-forested Floodplain | 0.36 | 120 | -0.20 | -0.11 | 0.09 |
| Temperate Non-Forested Weland | 0.34 | 120 | -0.22 | -0.10 | 0.13 |
| Temperate Forested Floodplain | 0.10 | 48 | -0.10 | -0.10 | 0.00 |
| Temperate Non-forested Floodplain | 0.10 | 45 | -0.04 | -0.02 | 0.01 |
| Wet Savanna | 0.16 | 59 | -0.27 | -0.21 | 0.06 |
| Salt Marsh | 0.09 | 35 | -0.04 | -0.01 | 0.02 |
| Mangroves | 0.12 | 38 | -0.39 | -0.32 | 0.07 |
| Temperate Savannas | 6.83 | 2921 | -3.17 | -2.63 | 0.54 |
| Temperate Evergreen Broadleaf | 3.33 | 1268 | -3.60 | -3.39 | 0.21 |
| Mediterranean Shrubland | 1.47 | 575 | -0.75 | -0.60 | 0.15 |
| Total | 133.56 | 59424 | -140.58 | -108.35 | 32.23 |









**Table 5.** Sensitivity of global CO consumption, net flux and production (units are Tg CO yr$^{-1}$) to changes in
atmospheric CO, soil organic carbon (SOC), precipitation (Prec) and air temperature (AT)

|  | Baseline | CO +30% | CO -30% | SOC +30% | SOC -30% | Prec +30% | Prec -30% | AT +3°C | AT -3°C |
|---|---|---|---|---|---|---|---|---|---|
| Consumption | -147.65 | -164.14 | -131.12 | -175.37 | -119.90 | -150.72 | -143.50 | -190.59 | -114.83 |
| Change (%) | 0.00 | -11.17 | 11.19 | -18.78 | 18.79 | -2.08 | 2.81 | -29.09 | 22.23 |
| Net flux | -113.65 | -130.15 | -97.12 | -131.18 | -96.10 | -116.97 | -109.32 | -144.23 | -89.58 |
| Change (%) | 0.00 | -14.51 | 14.54 | -15.42 | 15.44 | -2.92 | 3.81 | -26.90 | 21.18 |
| Production | 33.99 | 33.99 | 33.99 | 44.19 | 23.80 | 33.74 | 34.17 | 46.36 | 25.25 |
| Change (%) | 0.00 | 0.00 | 0.00 | 30.00 | -30.00 | -0.75 | 0.53 | 36.39 | -25.72 |
























**Table 6.** Effects of annual and monthly climate precipitation (Prec), air temperature (Tair), soil organic carbon
(SOC), soil temperature (Tsoil), soil moisture (Msoil) and atmospheric CO (CO air) on absolute values of
consumption, production and net flux for different regions and the globe during the 20th Century

| | | Monthly | | | | | Annual | | | | |
| --- | --- | --- | --- | --- | --- | --- | --- | --- | --- | --- | --- |
| | | North-45°N | 45°N-0° | 0°-45°S | 45°S-South | Global | North-45°N | 45°N-0° | 0°-45°S | 45°S-South | Global |
| Prec | Consumption | 0.91 | 0.96 | 0.92 | -0.34 | 0.87 | 0.65 | 0.21 | 0.26 | 0.13 | 0.52 |
| | Production | 0.91 | 0.70 | 0.45 | -0.34 | 0.82 | 0.63 | 0.10 | 0.15 | -0.11 | 0.47 |
| | Net flux | 0.91 | 0.97 | 0.94 | -0.33 | 0.87 | 0.65 | 0.25 | 0.31 | 0.32 | 0.54 |
| Tair | Consumption | 0.97 | 0.98 | 0.91 | 0.96 | 0.95 | 0.92 | 0.93 | 0.88 | 0.84 | 0.91 |
| | Production | 0.96 | 0.83 | 0.72 | 0.98 | 0.94 | 0.92 | 0.92 | 0.91 | 0.95 | 0.91 |
| | Net Flux | 0.97 | 0.97 | 0.88 | 0.90 | 0.95 | 0.91 | 0.92 | 0.85 | 0.62 | 0.91 |
| SOC | Consumption | -0.19 | 0.07 | 0.21 | -0.01 | 0.15 | 0.68 | 0.90 | 0.92 | 0.47 | 0.92 |
| | Production | -0.19 | 0.31 | 0.47 | -0.02 | 0.24 | 0.72 | 0.92 | 0.92 | 0.50 | 0.93 |
| | Net Flux | -0.19 | 0.03 | 0.14 | 0.00 | 0.13 | 0.67 | 0.88 | 0.91 | 0.38 | 0.91 |
| Tsoil | Consumption | 0.97 | 0.98 | 0.92 | 0.96 | 0.95 | 0.94 | 0.93 | 0.88 | 0.85 | 0.95 |
| | Production | 0.97 | 0.83 | 0.72 | 0.98 | 0.94 | 0.94 | 0.92 | 0.91 | 0.96 | 0.95 |
| | Net Flux | 0.98 | 0.97 | 0.88 | 0.90 | 0.95 | 0.93 | 0.93 | 0.86 | 0.63 | 0.95 |
| Msoil | Consumption | 0.85 | 0.96 | 0.92 | 0.19 | 0.76 | 0.03 | 0.22 | 0.14 | 0.26 | 0.22 |
| | Production | 0.85 | 0.75 | 0.44 | 0.14 | 0.69 | -0.02 | 0.12 | 0.02 | 0.05 | 0.17 |
| | Net Flux | 0.84 | 0.96 | 0.95 | 0.25 | 0.77 | 0.04 | 0.26 | 0.19 | 0.40 | 0.24 |
| CO Air | Consumption | -0.66 | -0.76 | -0.29 | 0.14 | -0.48 | 0.87 | 0.88 | 0.81 | 0.98 | 0.91 |
| | Production | -0.70 | -0.66 | 0.08 | -0.40 | -0.66 | -0.36 | -0.48 | -0.54 | -0.44 | -0.57 |
| | Net Flux | -0.64 | -0.73 | -0.35 | 0.55 | -0.41 | 0.92 | 0.91 | 0.88 | 0.99 | 0.94 |














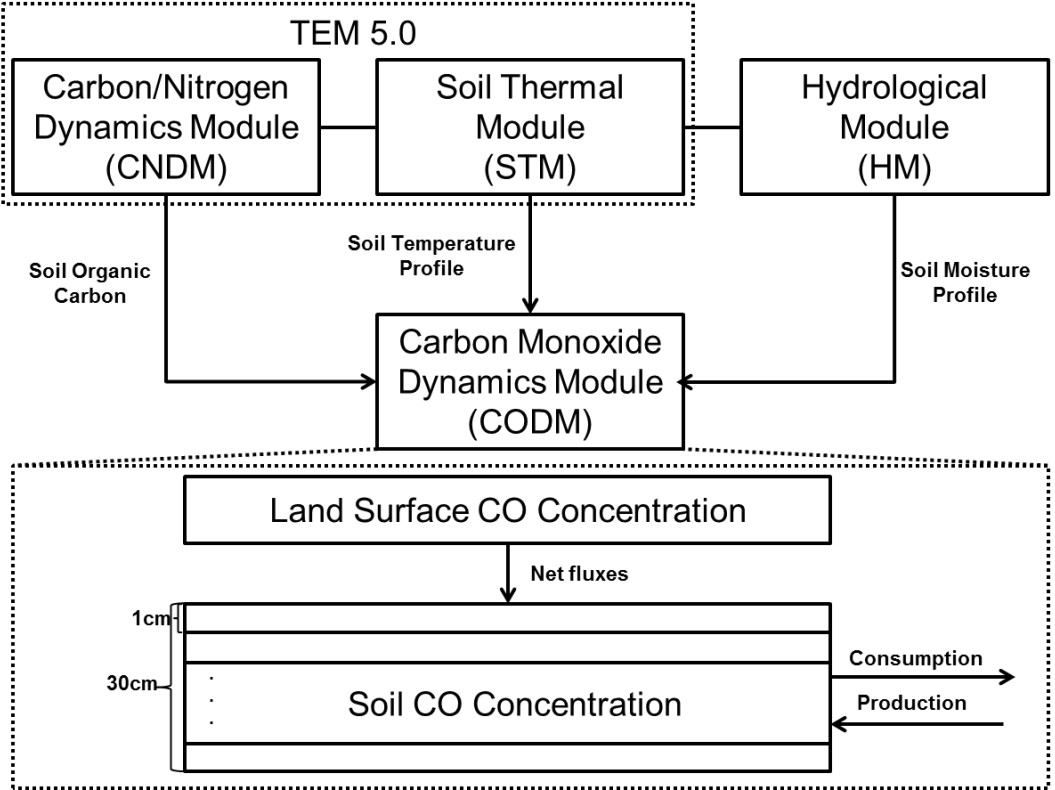


**Figure 1. The model framework** includes a carbon and nitrogen dynamics module (CNDM), a soil thermal module (STM) from Terrestrial Ecosystem Model (TEM) 5.0 (Zhuang et al., 2001, 2003), a hydrological module (HM) based on a Land Surface Module (Bonan, 1996; Zhuang et al., 2004), and a carbon monoxide dynamics module (CODM). The detailed structure of CODM includes land surface CO concentration as top boundary and thirty 1 cm thick layers (totally 30 cm) where consumption and production would happen inside.



































**Figure 2.** Evaluation of thermal and hydrology module at four sites: (a) Boreal Evergreen Needle Leaf Forests, (b)
Temperate Deciduous Broadleaf Forests. (1) shows the soil temperature comparison between model simulations (gray
line) and observations (black line) and (2) shows the soil moisture comparison between model simulations (gray line) and
observations (black line).









**Figure 2. Contd.** Evaluation of thermal and hydrology module at four sites: (c) Tropical Moist Forest, (d) Tropical Forest-Savanna. (1) shows the soil temperature comparison between model simulations (gray line) and observations (black line) and (2) shows the soil moisture comparison between model simulations (gray line) and observations (black line)







**Figure 3.** Parameter ensemble experiment results: Each parameter has 50 calibrated values generated from running SCE-UA-R 50 times independently. Parameters are normalized to their largest potential values described in Table 2. (a1) and (a2) are temperate coniferous forest normalized parameter distribution boxplots and CO flux comparisons between model simulations (solid line, using mean value of parameters) and observations ("+", red lines represent error bar), respectively. For each box, line top, box top, horizontal line inside box, box bottom and line bottom represent maximum, third quartile, median, first quartile and minimum of 50 parameter values. Red dot represents the mean value of 50 parameter values. (b1) and (b2) are plots for temperate deciduous forest; (c1) and (c2) are for boreal forest; (d1) and (d2) are for grassland. Grassland observation data is the sum of hourly observations so there is no error bar presented.





**Figure 3. Contd.** Parameter ensemble experiment results: Each parameter has 50 calibrated values generated from running SCE-UA-R 50 times independently. Parameters are normalized to their largest potential values described in Table 2. (a1) and (a2) are temperate coniferous forest normalized parameter distribution boxplots and CO flux comparisons between model simulations (solid line, using mean value of parameters) and observations ("+", red lines represent error bar), respectively. For each box, line top, box top, horizontal line inside box, box bottom and line bottom represent maximum, third quartile, median, first quartile and minimum of 50 parameter values. Red dot represents the mean value of 50 parameter values. (b1) and (b2) are plots for temperate deciduous forest; (c1) and (c2) are for boreal forest; (d1) and (d2) are for grassland. Grassland observation data is the sum of hourly observations so there is no error bar presented.



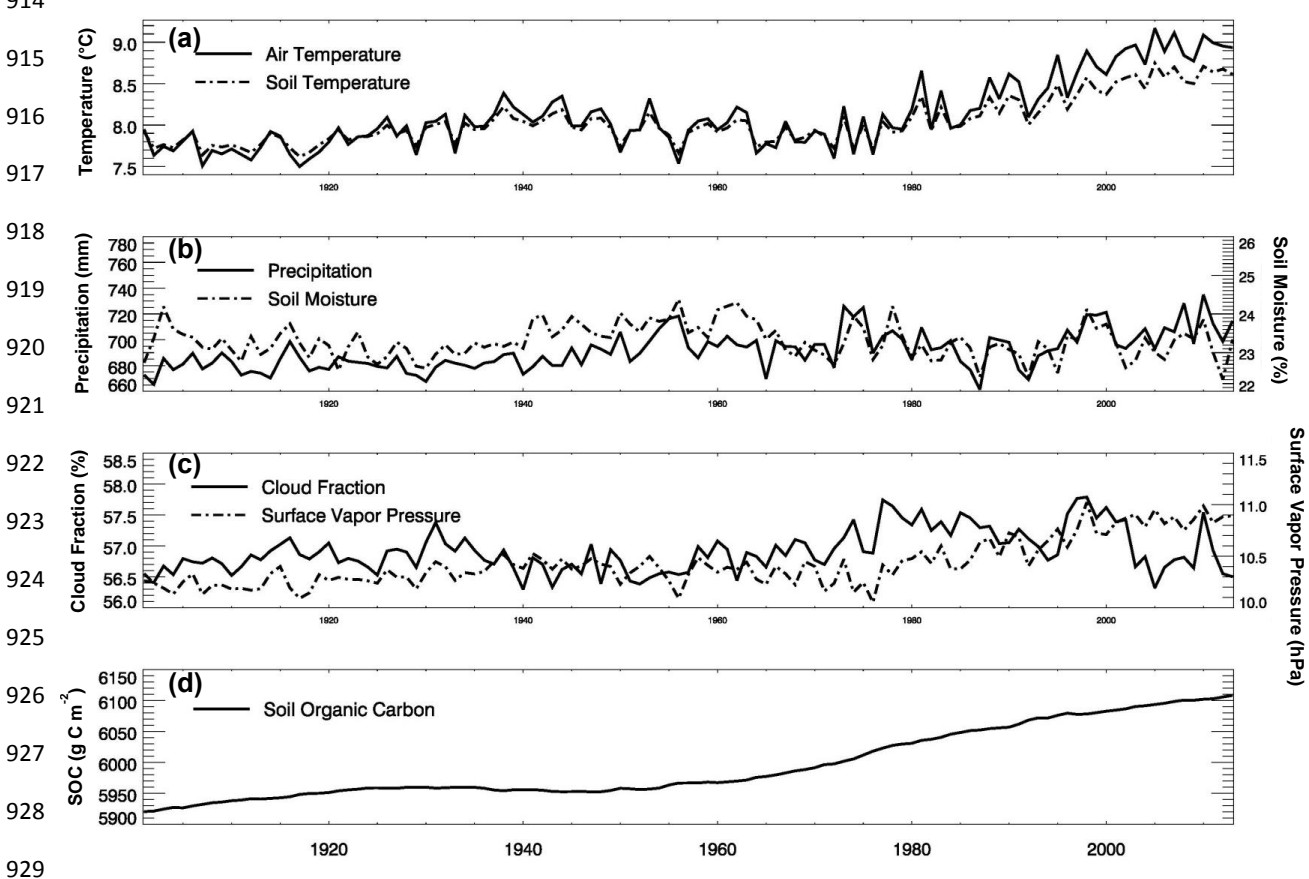

**Figure 4.** Historical global land surface mean climate, and simulated global mean soil moisture, soil temperature and SOC for the period 1901-2013.


**Figure 5.** CO surface concentration data from MOPITT satellite (ppbv): (a) global mean CO surface concentrations from MOPITT during 2000-2013; (b) the CO annual surface concentrations from both MOPITT and empirical functions (Potter et al., 1996).






**Figure 6.** Global mean climate from RCP2.6, RCP4.5 and RCP8.5 data sets and simulated global mean soil temperature, moisture and SOC: (a)-(g) are land surface air temperature (°C), soil temperature (°C), precipitation (mm), soil moisture (%), surface water vapor pressure (hpa), cloud fraction (%), and SOC (mg m$^{-2}$), respectively.



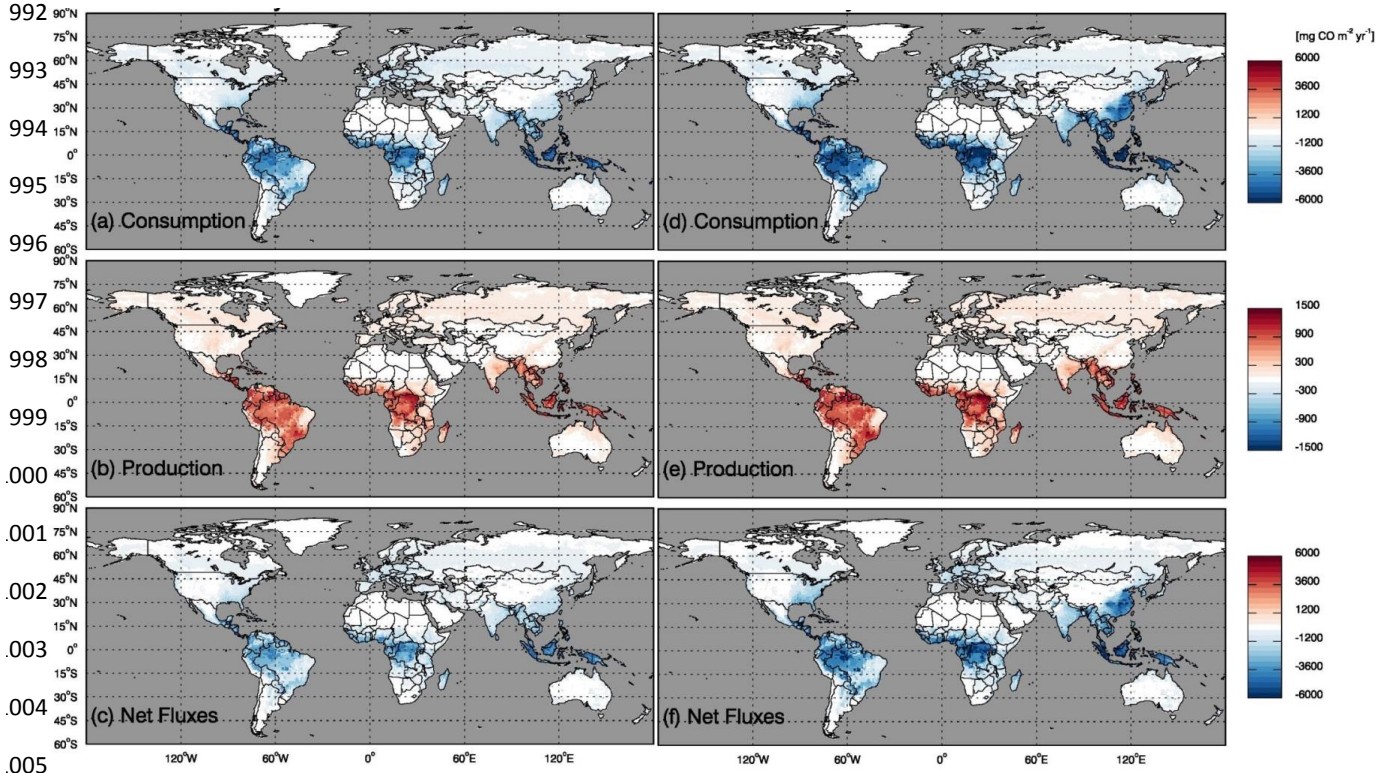

**Figure 7.** Global annual mean soil CO fluxes (mg CO m$^{-2}$ yr$^{-1}$) during 1901-2013, estimated using constant CO concentration data (left side) and mean annual global soil CO fluxes during 2000-2013 using MOPITT CO atmospheric surface concentration data (right side)





**Figure 8.** Global mean CO consumption, production and net flux(Tg CO yr$^{-1}$): (a) from 1901 to 2013, estimated with constant CO surface concentration data and (b) from 2000 to 2013 with MOPITT CO surface concentration data.





**Figure 9.** Global annual mean latitudinal distributions of soil CO consumption, production and net flux: (a) during 1901-2013 (Tg CO yr$^{-1}$) estimated with constant CO surface concentration data and (b) during 2000-2013 estimated with MOPITT CO surface concentration data.





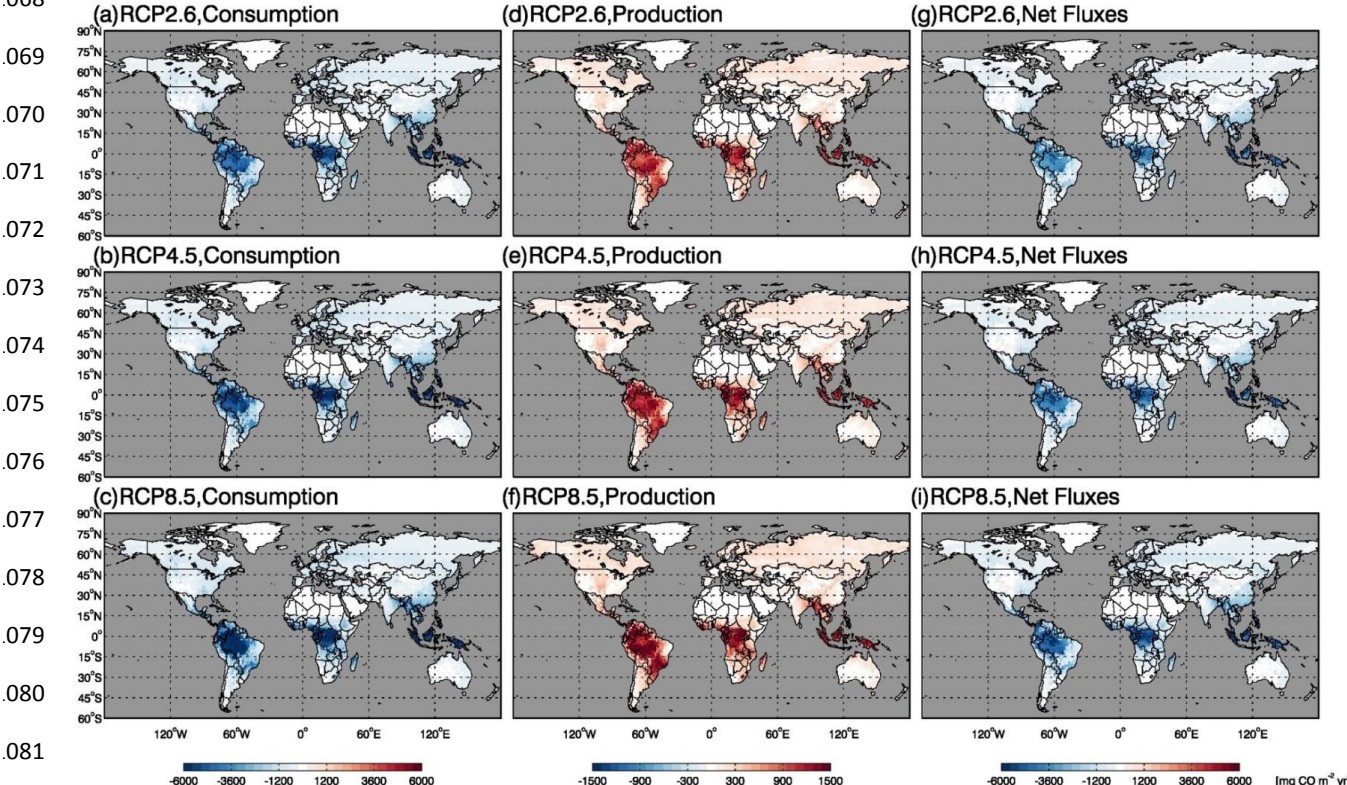

**Figure 10.** Global annual mean CO consumption, production and net flux (mg CO m$^{-2}$ yr$^{-1}$) under future climate scenarios RCP2.6, RCP4.5 and RCP8.5 during 2014-2100



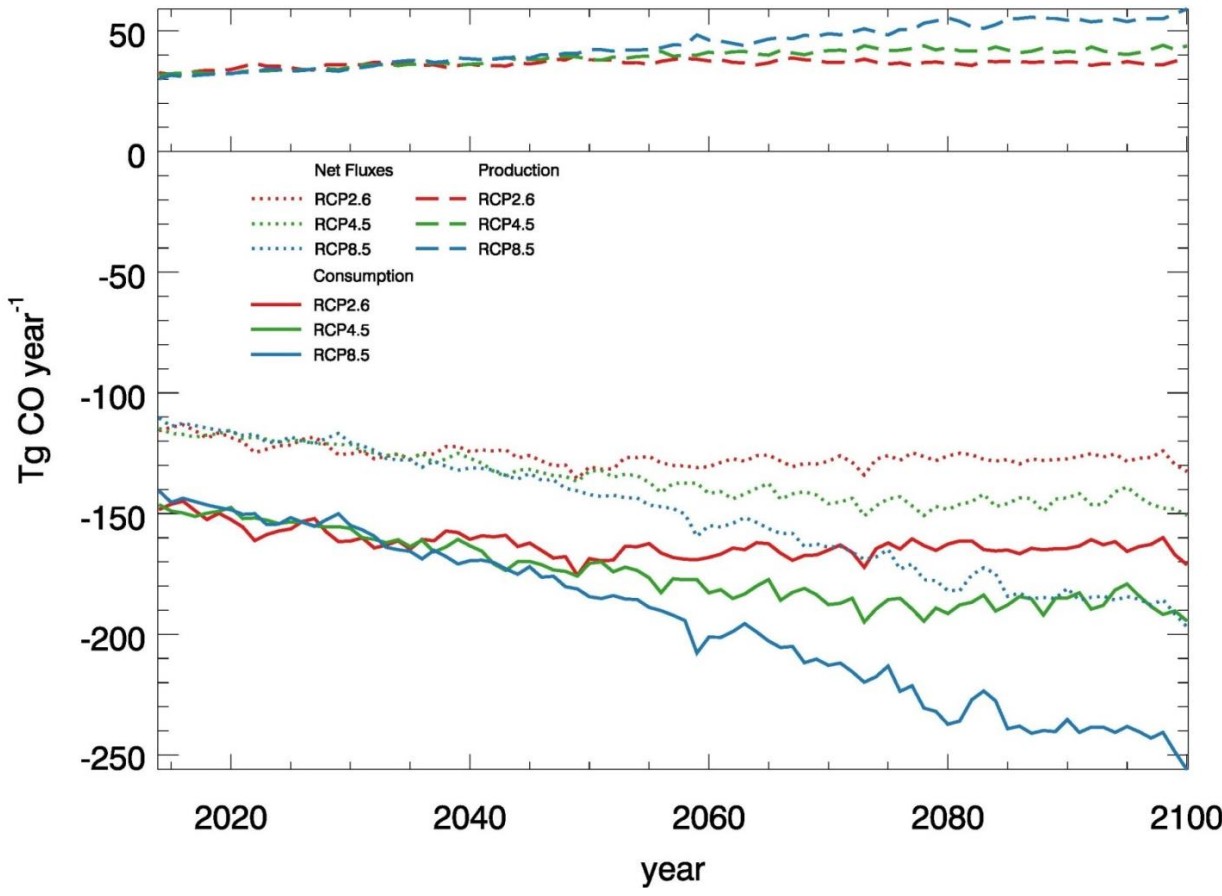

.085

**Figure 11.** Future Global mean soil CO consumption, net flux and production (Tg CO yr$^{-1}$) under future climate scenarios
RCP2.6, RCP4.5 and RCP8.5 during 2014-2100



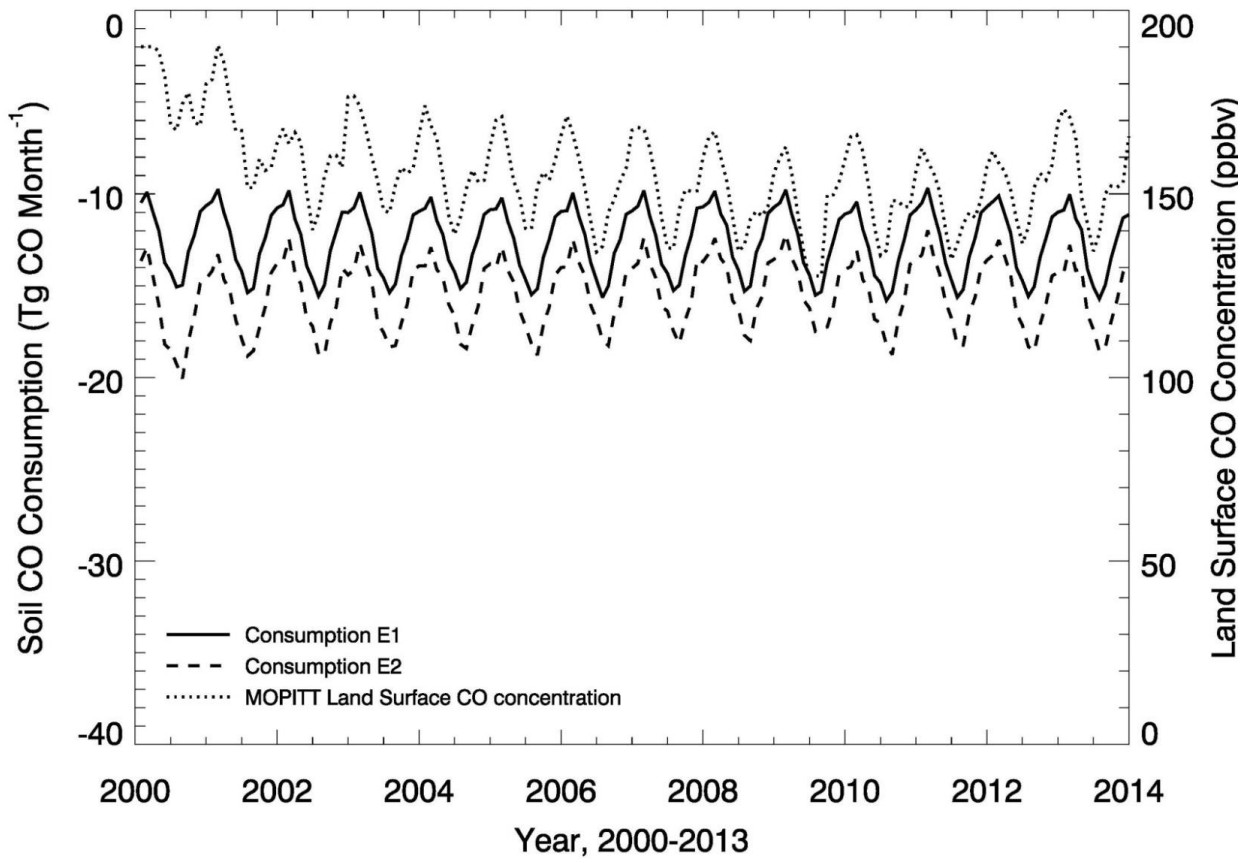

.088

.089 **Figure 12.** Monthly time series of MOPITT atmospheric CO concentration (ppbv) and soil CO consumption from model

.090 simulations E1 and E2 (Tg CO mon$^{-1}$)