# Peer review of "An Analysis Using a Process-Based Biogeochemistry Model"

_Atmospheric Chemistry and Physics, 2017_

## Referee Comment (RC1) · S. Yonemura (Referee) · 18 Oct 2017

At first, as authors submitted this manuscript to ACP, I consider that ACP is more appropriate journal than Biogeosciences for the publication of this study because results of this study can highly contribute to Atmospheric Chemistry.

General comments Most of the global estimation of CO soil consumption from previous studies is simple estimates of sum using deposition velocity and there is no sophisticated model study with biogeochemistry. After a long break of 15 years, authors did their best to estimate global soil CO consumption and production with extensive compilation of previous studies and with IPCC scenarios. So, the most advanced estimation

of the global soil CO consumption/production is shown in this study. This study summarized almost all the previous studies about soil CO processes. So, this study is important as a review paper. I recommend "accept" of this manuscript in ACP after a minor revision. Hereafter, I point some comments.

Specific comments Abstract The time step of the simulation in this study is monthly. Time step of the calculation is very important. So, please explicit describe about the time step of the calculation even in abstract.

Page 2 line 31 I do not think that this sentence is necessary.

Page 5 line 118 are in -> is

Page 5 line 127 withan-> with an

Page 6 line 139 Please spell out TEM at the first place.

Page 6 line 157 Vertical grid of 1cm can be used for simulation of $CO_2$ and $CH_4$ diffusion processes in soil, but, I consider that 1cm is still not so finely-gridded for the simulation to model soil CO consumption because of rapid CO consumption in soil (especially very active soil to consume CO). Some soils are strong consumers of CO and these soils absorb CO within 2-3cm of top soil layer. Though this comment does not deny robustness of the results of this study, I recommend that these technical aspects should be mentioned in the Discussion section (as in line 376) to be kind for readers who may study soil CO consumption. Furthermore, authors properly used implicit (Crank-Nicolson) method in order to be independent from time-step which must be set as short as possible in case of explicit method because the vertical grid must be finer for soil CO consumption for explicit method.

Page7 line 160 "$i$" and "$t$" should be italic.

Page 12 line 297 Figure 3 (a2,b2,c2,d2) I cannot understand which observations in Table 1 were plotted in the sub-figures.

Page 12 line 305- Page 13 Direction of consumption and production or net flux is misleading. I felt that minus expression of values is difficult to see through.

Page 16 Model Uncertainties and Limitations I consider that CO concentrations at soil surface (environmental CO concentration for soil) is a little different from CO data from MOPITT. Boundary-layer processes are also complex. A comment about this point is necessary.

Table2-6 Values were centered but should be formatted to be easily understood. For example, Earea/R (K) in Table 2 In place of 8801 14165, 8801 14165 is better.

Figure2 Units of soil moisture are different among (a2), (b2) (c2) and (d2). My concern is about the highness of the volumetric soil moisture. The soil moistures in (a2) (d2) are too high? Volumetric soil moisture contents (a2) are too high, as high as 80%. Normal soils have no capacity to hold such high moistures. The units of the soil moisture contents are all volumetric (m3/m3) ? Please check and if the shown volumetric soil moistures are correct, please mention reasons.

Figure 3 (c2) and (d2) Why authors showed over-scale (CO emission) graph? The y-axis of (c2) should be -10 to 2

Figure 4 Please write clear the meaning of "global land surface". Global land surface includes Antarctic area? Normal readers think that global average temperature is about 15C but the shown temperature (a) is between 7.5-9C.

Figure 6 Why SOC increases sharply before 2100?

―――――――――――――――――――――――

---

## Referee Comment (RC2) · Anonymous Referee #2 · 3 Dec 2017

**Review of "Global Soil Consumption of Atmospheric 1 Carbon Monoxide: An Analysis Using a 2 Process-Based Biogeochemistry Model" by Liu et al.**

The paper presents a bottom-up model for the exchange of CO between soil and atmosphere. The CO soil exchange module is process-based and includes explicit gross soil consumption and production fluxes. This module uses soil moisture and temperature, and atmospheric CO mole fraction data.

The work has several main components: (1) validation of the model using data from four field measurement stations; (2) simulation of the past using a constant distribution of atmospheric CO; (3) simulation of the recent period using MOPITT data, and (4) simulation of the future using the same constant distribution from point (2). Additionally, sensitivity tests are used for determining the influence of various parameters on soil fluxes.

I find the paper interesting, and the subject fits very well into the ACP area of interest. However, I find the paper quite descriptive – I miss some discussion and conclusion on the meaning and significance of the results, and some outlook on the further use of this model. The paper is not very well organized, with some parts difficult to follow, and at some places I found it difficult to extract the relevant information from the text.

I recommend publication after the following points are addressed.
I include below only those issues from the access review that have not been solved.

**Main concern: CO concentration and flux mismatch between MOPITT and constant function simulations**
The CO soil deposition is largest flux and is strongly dependent on the atmospheric CO mole fraction. The constant CO mole fractions used in past and future model runs are not realistic, at least for the period 2000-2013, as shown by the large mismatch in magnitude with the MOPITT satellite data for the overlapping period (Fig . 5b).
Correspondingly, the annual fluxes calculated over 1901-2013 using the constant CO distribution (determined by the function shown at lines 263) are very different from the fluxes calculated using the CO MOPITT data for the shorter period 2000-2013.
The authors report both sets of incompatible fluxes as results. Assuming that the satellite measured values are close to the truth, then the fluxes calculated using the constant function, for the 20[th] and 21[st] centuries, are obviously wrong. However, these are reported as main results and the authors claim to "quantify global soil budget for the 20[th] and 21[st] centuries".

In my opinion these fluxes should not be reported in the actual form. They can however be used to study the variability and the relative magnitude of the components, and the relative variation in time. If the exercise is only for understanding the controls and the relative evolution (as the authors state in their reply to reviewer 2), this can be accepted as long as the limitations are made clear and the fluxes are not claimed to be solved.

Here are some ideas on what can be done:
- scale the CO distribution function in such a way that it matches the satellite observations for the overlapping period. This would need an assumption for the temporal evolution of CO during the 20[th] century

- report the results not in terms of fluxes, but in terms of net deposition velocities. This would require changes through the paper.
- keep the fluxes as they are, but do not discuss the absolute values of the fluxes but only the relative variations

Besides these, the authors should be careful with claiming that they quantify the soil budget for the 20$^{th}$ and 21$^{st}$ centuries.
This issue should be discussed thoroughly in Sect. 4.3.

**Other general comments**
- Abstract: please report the same values given in the paper, either averages or ranges.
- please try to organize the paper in smaller paragraphs to improve readability
- I find the Introduction up to line 60 difficult to read, because of the long list of references following each bit of information. Please consider rephrasing and reducing the references that keep being repeated.
- The summary of the experiments is given both in 2.1 and 2.5, but the two descriptions do not seem to match. In 2.1. it is mentioned that the purpose was to "investigate the impact of …atmospheric CO concentrations " but the sensitivity tests that are meant for this are not mentioned here.
- is Michaelis-Menten kinetics necessary? The net uptake means in principle that at least the uppermost layer of soil has lower concentration than the atmosphere, and this is much lower than kCO. Are the CO concentrations in the lower soil layer much larger, and if yes, why? Related to this, please show the CO soil profiles for some typical situations.
-  please number the figures in the order they are discussed in text (e.g. Figs 7, 8) – or change the text to mention the figures in the right order.
- Section 4.2 should be reorganized and at least partly moved to results – see specific comments
- Section 4.3 discussed the model uncertainties and limitations, but ignores two of the major issues: (1) the CO concentrations mismatch between satellite and constant function, and (2) the overestimation of soil consumption at high temperatures.

**Specific comments**
- line 16: "constant spatially distributed" – can be understood wrongly as constant in space, which is not; I suggest to change to " constant in time, spatially distributed"
- line 20: "the largest sinks at 93 Tg CO yr-1" – is this from the 20$^{th}$ century or 2000-2013 simulation? Please specify. Same for the next phrase.
- lines 63 and 68: use the same units for the deposition velocity
- lines 61-68: the phrase is unclear, consider breaking it into smaller pieces. Also, quite some information is given on the studies listed, but not enough to actually understand what they did and what the differences are. Please consider either giving more details, or shortening this part.
- lines 62 – 66: the text seems contradictory: the uptake flux when using a constant deposition velocity globally (115-230 Tg) is smaller than when using the same deposition velocity in general  and some area set to zero (300 Tg)  - please explain or reformulate. "With different approaches" is too unspecific and does not really say anything.
- line 67: " using empirical approaches with higher probability for lower values" – unclear, higher probability of what? lower values for what?
- lines 68 – 70: which other substances? what are other deposition velocities? please give examples if you mention this.

- lines 75 – 91: a lot of this discussion on the thermal and photo degradation is irrelevant for this paper, especially the part on photo degradation which is not included in the model. Please reduce the irrelevant parts.
- lines 92 – 93: it is unclear to me what this phrase means. I think what the authors may intend to say is that *little attention has been paid to CO (including soil consumption and production) in global (chemistry?) models.* Please reformulate if true.
- lines 97-98: suggest replacing "oxidation from soil bacteria and microbes" by "oxidation by soil microbes". Bacteria are microbes.
- line 110: CO emission is abiotic, right?
- line 156: "determined by the mass balance" – unclear what this means: which mass balance, of what, between what? please clarify.
- line 162: this term represents all the consumption; remove "due to oxidation"
- line 171: "modeled as an anaerobic process" there is nothing in Eq. 2 that makes it anaerobic
- lines 186- 188: phrase unclear - I think the authors mean that Eq 2.2 will overestimate CO consumption because in reality CO consumption decreases at high temperatures, while in Eq 2.2 CO keeps increasing with temperature. Please reformulate.
- line 193: i do not understand what Pr(t,i) is
- line 198: should it be 20 cm SOC?
- lines 275 – 277: give some details on the scenarios and datasets
- line 297: is the reported correlation really $r$, or is it $r^2$? Also, a correlation coefficient $r$ of 0.5 is usually not considered high correlation.
- lines 299 - 300: please compare the RMSEs reported to the CO fluxes, in order to give an impression on the relative errors.
- lines 318 – 319: "consume 42% and 58% of the total consumption, and produce 41% and 59% of total production" – please reformulate
- line 328: Table 3 does not show the deposition flux in mg/m2 day
- lines 329 – 343: I find the text in this paragraph somewhat misleading. The fluxes are presented as changing or increasing relative to the simulation for 20[th] century, which suggests a temporal evolution, but in fact they are different mainly because of a different model setup, i.e different atmospheric CO concentrations. Consider using "different" and "larger" instead of "increasing".
- line 359: "the rate ranges of increasing of consumption…" – I think it should be something like "the ranges of the rates of increase in consumption…"; please explain what these ranges are, are they corresponding to the three scenarios?
- line 382: the references should be given in the method section, not here
- Sect 4.2: The information here belongs mostly to Results, please reorganize. Also, the section is hard to follow – there are many correlations mentioned without much coherence. Consider reformulating into a more focused way, with one paragraph per idea (e.g on annual time scales, the CO uptake is mostly correlated to X, Y, Z… and then comment more if needed on X ). Try to separate the annual and monthly results.
- lines 398 – 400 and Table 5: The effect of the SOC on the gross uptake flux seems too large. In my understanding, the text tries to explain that SOC increases the gross production which makes more CO available, which in turn leads to an increase in the gross CO consumption. But, for an increase in SOC of 30%, the production increases by about 10Tg/year, and the consumption increases by 28 Tg/year! How can that be? Where are the extra 18 Tg/year coming from?
- line 406: "as CO flux" – do you mean "as does the CO flux"?
- line 415: what is 0.91? Is it r or r^2, or something else? The same for line 418.
- line 425: the same data limitation is when using any method, not only SCE-UA-R, correct? If yes, remove "using SCE-UA-R method"

- line 427: "with RMSE … day-1" – I think this info has no meaning here
- Table 6: what are the numbers? what are the units? These are not absolute values of fluxes.
- line 796: I suggest to replace "would happen inside" by "take place"
- Fig. 2: is this really volumetric soil moisture? The values do not seem realistic; I think typical values for water holding capacity for most soils are around 50 % and that would give the saturation. Please check the units.
- Fig. 2-d2: please use the same units as in a2, b2 and c2 figures.

**Text comments**
- line 60: should be "… consumption to be …"
- line 81: "formations" should be "formation"
- line 118: "are in Pihlatie" should be "is Pihlatie"
- line 219: I think "misfit" should be "mismatch"
- line 266 and through the paper: "transient" is used wrongly. If what is meant is "variable" then please do use "variable". Transient does not mean variable, but something that disappears.
- Fig. 3: axes text too small; markers not visible in all figures (especially c2) , please consider using color markers
- Fig. 3 part 1, page 34: Remove from caption the explanation for c and d
- Fig. 3 part 2, page 35: Remove from caption the explanation for a and b
- Fig. 6: x labels not visible; some of the y labels cut
- Fig. 8: typo in legend: "producion"
- Fig. 9: typo in legend: "producion"

---

## Author Comment (AC1) · 11 Feb 2018

Author's response to referee 1:

Thank you very much for your supportive and precious comments! You helped us significantly improve this study.

1) Comments: Abstract The time step of the simulation in this study is monthly. Time step of the calculation is very important. So, please explicit describe about the time step of the calculation even in abstract.

Response: We have added the information in Abstract: "We develop a process-based biogeochemistry model to quantify CO exchange between soils and the atmosphere with a 5-minute internal time step at the global scale. The model is parameterized using CO flux data from the field and laboratory experiments for eleven representative ecosystem types. The model is then extrapolated to the global terrestrial ecosystems using monthly climate forcing data." From line 10 to line 15.

2) Comments: Page 2 line 31 I do not think that this sentence is necessary.

**Response: Deleted.**

3) Comments: Page 5 line 118 are in -> is

Response: Corrected. "The first study to report long-term and continuous field measurements of CO flux over grasslands using a micrometeorological eddy covariance (EC) method is Pihlatie et al. (2016)." From line 93 to line 95.

4) Comments: Page 5 line 127 withan-> with an

Response: Corrected. "A set of century-long simulations of 1901-2100 were also conducted using the atmospheric CO concentrations estimated with an empirical function (Badr & Probert, 1994; Potter et al., 1996)". From line

5) Comments: Page 6 line 139 Please spell out TEM at the first place.

Response: We have mentioned at the first place of introduction. "To improve the quantification of the global soil CO budget for the period 2000-2013 and CO deposition velocity for the 20th and 21st

centuries, this study developed a CO dynamics module (CODM) embedded in a process-based biogeochemistry model, the Terrestrial Ecosystem Model (TEM) (Zhuang et al., 2003, 2004, 2007)." From line 96 to line 99.

6) Comments: Page 6 line 157 Vertical grid of 1cm can be used for simulation of CO2 and CH4 diffusion processes in soil, but, I consider that 1cm is still not so finely-gridded for the simulation to model soil CO consumption because of rapid CO consumption in soil (especially very active soil to consume CO). Some soils are strong consumers of CO and these soils absorb CO within 2-3cm of top soil layer. Though this comment does not deny robustness of the results of this study, I recommend that these technical aspects should be mentioned in the Discussion section (as in line 376) to be kind for readers who may study soil CO consumption. Furthermore, authors properly used implicit (Crank-Nicolson) method in order to be independent from time-step which must be set as short as possible in case of explicit method because the vertical grid must be finer for soil CO consumption for explicit method.

Response: Thanks much for your suggestions. In this revision, we have tested the model using 3, 15, 30, 300, 3000 thin layers to examine the influence of layer thickness. It turned out that we have chosen the proper layers division and more layers will need much more computing time, but not show further improvement. We have summarized these tests in Figure 12 and Section 4.3, line 452 to 460.

7) Comments: Page7 line 160 "i" and "t" should be italic.

Response: Corrected. "Where C(t, i) is the CO concentration in layer i and at time t, units are mg m- 3." Line 142.

8) Comments: Page 12 line 297 Figure 3 (a2,b2,c2,d2) I cannot understand which observations in Table 1 were plotted in the sub-figures.

**Response: We added information to indicate the site being used in Figure 3 caption.**

9) Comments: Page 12 line 305- Page 13 Direction of consumption and production or net flux is misleading. I felt that minus expression of values is difficult to see through.

**Response: Thanks for pointing this out. We have changed all values presented as ranges like "From - 180 to -197, 34 to 36 and -145 to -163 Tg CO yr-1".**

10) Comments: Page 16 Model Uncertainties and Limitations I consider that CO concentrations at soil surface (environmental CO concentration for soil) is a little different from CO data from MOPITT. Boundary-layer processes are also complex. A comment about this point is necessary.

Response: We have revised Section 4.3 to address your comments. ". Third, the derived CO surface concentration is lower than MOPITT CO surface concentration, which will lead to overestimation of CO deposition velocity during 1901-2100." From Line 450 to 452.

11) Comments: Table2-6 Values were centered but should be formatted to be easily understood. For example, Earea/R (K) in Table 2 In place of 8801 14165, 8801 14165 is better.

**Response: We have now centered all values and names of parameters.**

12) Comments: Figure2 Units of soil moisture are different among (a2), (b2) (c2) and (d2). My concern is about the highness of the volumetric soil moisture. The soil moistures in (a2) (d2) are too high? Volumetric soil moisture contents (a2) are too high, as high as 80%. Normal soils have no capacity to hold such high moistures. The units of the soil moisture contents are all volumetric (m3/m3) ? Please check and if the shown volumetric soil moistures are correct, please mention reasons.

Response: Thank you for pointing this out. In this revision, we traced back to Nakai et al. (2013) and found that our units and values are the same as they presented. The reason why the values were so high is that the volumetric soil moisture (VSM) was converted from the water content reflectometry (WCR) probe output period using an empirical calibration function of Bourgeau-Chavez et al. (2012) for 5cm-30cm layer. Although Bourgeau-Chavez et al. (2012) provided calibration functions for each soil horizon (i.e., dead moss, upper duff, lower duff, and mineral soil), some of them resulted in values greater than 100% VSM in Nakai et al. (2013) study. The model estimated high VSM (close to 80%) is due to top 10 cm moss in the model which has a saturation VSM of 0.8. We added the discussion on Figure 2 caption in this revision. From line 482 to 485.

13) Comments: Figure 3 (c2) and (d2) Why authors showed over-scale (CO emission) graph? The y-axis of

(c2) should be -10 to 2

**Response: We have corrected the Y-axis's range to -10 to 2 in Figure 3 (c2).**

14) Comments: Figure 4 Please write clear the meaning of "global land surface". Global land surface includes Antarctic area? Normal readers think that global average temperature is about 15C but the shown temperature (a) is between 7.5-9C.

Response: We have added extra information in caption of Figure 4 and Figure 6: "Global land surface (excluding Antarctic area and ocean area)"

15) Comments: Figure 6 Why SOC increases sharply before 2100?

Response: In this revision, we have fixed this problem. The fixed values of SOC is showed in figure 6.

We also rerun the model to remove the influence of odd SOC to future prediction of CO dynamics.

---

## Author Comment (AC2) · 11 Feb 2018

Author's response to referee 2:

**Thank you for your constructive comments, which helped us improve this study. Here are our responses to each of your comments:**

1) Main Concern: CO concentration and flux mismatch between MOPITT and constant function simulations The CO soil deposition is largest flux and is strongly dependent on the atmospheric CO mole fraction. The constant CO mole fractions used in past and future model runs are not realistic, at least for the period 2000-2013, as shown by the large mismatch in magnitude with the MOPITT satellite data for the overlapping period (Fig . 5b).

Correspondingly, the annual fluxes calculated over 1901-2013 using the constant CO distribution (determined by the function shown at lines 263) are very different from the fluxes calculated using the CO MOPITT data for the shorter period 2000-2013.

The authors report both sets of incompatible fluxes as results. Assuming that the satellite measured values are close to the truth, then the fluxes calculated using the constant function, for the 20th and 21st centuries, are obviously wrong. However, these are reported as main results and the authors claim to "quantify global soil budget for the 20th and 21st centuries".

In my opinion these fluxes should not be reported in the actual form. They can however be used to study the variability and the relative magnitude of the components, and the relative variation in time.

If the exercise is only for understanding the controls and the relative evolution (as the authors state in their reply to reviewer 2), this can be accepted as long as the limitations are made clear and the fluxes are not claimed to be solved.

Here are some ideas on what can be done:

- scale the CO distribution function in such a way that it matches the satellite observations for the overlapping period. This would need an assumption for the temporal evolution of CO during the 20th century

- report the results not in terms of fluxes, but in terms of net deposition velocities. This would require changes through the paper.

- keep the fluxes as they are, but do not discuss the absolute values of the fluxes but only the relative variations

Besides these, the authors should be careful with claiming that they quantify the soil budget for the 20th and 21st centuries.

This issue should be discussed thoroughly in Sect. 4.3.

**Response: We highly appreciate your constructive suggestions and comments.  We have chosen your second recommendation and changed historical simulation (1901-2013) and future simulation (2014-2100) results to be presented as deposition velocity by using the method in Seinfeld, et al. (1998). In this revision, what we have done include: 1) We only discussed absolute values of simulations during the period 2000-2013 when using MOPITT CO surface concentration data; 2) We removed all parts that involve absolute values of CO consumption, production and net fluxes; 3) We have plotted out new figures to present deposition velocity for 1901-2100; 4) We have re-organized the paper with a clearer experiment orders: E1: simulations during the period 2000-2013 using MOPITT CO surface concentration data; E2: simulations during the period 1901-2100 using constant in time, spatially distributed CO surface concentration data; E3: sensitivity tests; and 5) We added discussion of CO concentration influences on CO dynamics in Section 4.3, line 450 to line  452.**

**Other general comments:**

2) - Abstract: please report the same values given in the paper, either averages or ranges.

**Response: We have used ranges now throughout the paper.**

3) - please try to organize the paper in smaller paragraphs to improve readability

**Response: We have reorganized a number of paragraphs to get them smaller. In addition, we have also reduced many repeated references in Introduction.**

4) - I find the Introduction up to line 60 difficult to read, because of the long list of references following each bit of information. Please consider rephrasing and reducing the references that keep being repeated.

**Response: We have significantly revised the Introduction section and also removed repeated or non-necessary references. Please see line 33 to line 52.**

5) - The summary of the experiments is given both in 2.1 and 2.5, but the two descriptions do not seem to match. In 2.1. it is mentioned that the purpose was to "investigate the impact of …atmospheric CO concentrations "but the sensitivity tests that are meant for this are not mentioned here.

**Response: We have corrected this and now they describe the same experiments. See Section 2.1 and Section 2.5.**

6) - is Michaelis-Menten kinetics necessary? The net uptake means in principle that at least the uppermost layer of soil has lower concentration than the atmosphere, and this is much lower than kCO. Are the CO concentrations in the lower soil layer much larger, and if yes, why? Related to this, please show the CO soil profiles for some typical situations.

**Response: In this revision, we have re-considered the Michaelis-Menten kinetics and presented soil CO concentration profile in figure 12. We found that the Michaelis-Menten kinetics can't be replaced in our model since the soil CO concentration can be bigger than atmospheric CO surface concentration in the days of net emissions (Figure 12f). Replacing it with a simple version will overestimate the oxidation rate when soil CO concentration is comparable with atmospheric CO concentration. We have discussed this in Section 4.3. From line 460 to line 465**

7) - please number the figures in the order they are discussed in text (e.g. Figs 7, 8) – or change the text to mention the figures in the right order.

**Response: We have carefully checked the order of figures and their uses in main text.**

8) - Section 4.2 should be reorganized and at least partly moved to results – see specific comments

**Response: We have moved sensitivity test results to Section 3.4 and we have reorganized 4.2 based on specific comments. See section 3.4 from line 361 to line 369, and section 4.2 from line 397 to line 408.**

9) - Section 4.3 discussed the model uncertainties and limitations, but ignores two of the major issues: (1) the CO concentrations mismatch between satellite and constant function, and (2) the overestimation of soil consumption at high temperatures.

**Response: We have added these two points in Section 4.3 in this revision. See line 450 to 452 and line 441 to line 443 for issue 1) and issue 2), respectively.**

**Specific comments**

10) - line 16: "constant spatially distributed" – can be understood wrongly as constant in space, which is not; I suggest to change to " constant in time, spatially distributed"

**Response: We have rephrased it following your suggestion. "By assuming that the spatially-distributed atmospheric CO concentrations (~128 ppbv) are not changing over time". From line 21 to line 23.**

11) - line 20: "the largest sinks at 93 Tg CO yr-1" – is this from the $20_{th}$ century or 2000-2013 simulation? Please specify. Same for the next phrase.

**Response: This value is for 2000-2013 simulation. We corrected. See line 19.**

12) - lines 63 and 68: use the same units for the deposition velocity

**Response: Corrected. Now units for deposition velocity are mm $s^{-1}$ throughout the paper manuscript.**

13) - lines 61-68: the phrase is unclear, consider breaking it into smaller pieces. Also, quite some information is given on the studies listed, but not enough to actually understand what they did and what the differences are. Please consider either giving more details, or shortening this part.

**Response: We have shortened this part to increase readability and avoid misleading. See line 53 to line 57.**

14) - lines 62 – 66: the text seems contradictory: the uptake flux when using a constant deposition velocity globally (115-230 Tg) is smaller than when using the same deposition velocity in general and some area set to zero (300 Tg) - please explain or reformulate. "With different approaches" is too unspecific and does not really say anything.

**Response: We have removed this for shortening the paragraph and reducing confusing. See line 57 to line 59.**

15) - line 67: " using empirical approaches with higher probability for lower values" – unclear, higher probability of what? lower values for what?

**Response: This part has been removed in this revision. See line 57 to line 59.**

16) - lines 68 – 70: which other substances? what are other deposition velocities? please give examples if you mention this.

**Response: We have rewritten this sentence and now just presented the ranges for all vegetation types. Detailed deposition velocity for different vegetation types can be found in King (1999a) and Castellanos et al. (2011). See line 57 to line 59.**

17) - lines 75 – 91: a lot of this discussion on the thermal and photo degradation is irrelevant for this paper, especially the part on photo degradation which is not included in the model. Please reduce the irrelevant parts.

**Response: We have reduced the part related to photo-degradation. Now we only mentioned what is photo-degradation. See line 63 to line 64.**

18) - lines 92 – 93: it is unclear to me what this phrase means. I think what the authors may intend to say is that little attention has been paid to CO (including soil consumption and production) in global (chemistry?) models. Please reformulate if true.

**Response: We have removed the sentence. See line 70.**

19) - lines 97-98: suggest replacing "oxidation from soil bacteria and microbes" by "oxidation by soil microbes". Bacteria are microbes.

**Response: Corrected. See line 74 to line 75.**

20) - line 110: CO emission is abiotic, right?

**Response: We have removed the "CO emission" from the sentence. "One reason is that there is an incomplete understanding of biological processes of uptake". See line 85 to line 86.**

21) - line 156: "determined by the mass balance" – unclear what this means: which mass balance, of what, between what? please clarify.

**Response: We have added the detailed description: "Net exchange of CO between the atmosphere and soil is determined by the mass balance approach (net flux = total production – total oxidation – total soil CO concentration change)."**

22) - line 162: this term represents all the consumption; remove "due to oxidation"

**Response: Removed "due to oxidation" from the sentence. See line 144.**

23) - line 171: "modeled as an anaerobic process" there is nothing in Eq. 2 that makes it anaerobic

**Response: We have rewritten the sentence and removed "modeled as an anaerobic process" from description: "CO consumption is modeled in unsaturated soil pores", see line 153**

24) - lines 186- 188: phrase unclear - I think the authors mean that Eq 2.2 will overestimate CO consumption because in reality CO consumption decreases at high temperatures, while in Eq 2.2 CO keeps increasing with temperature. Please reformulate.

**Response: We have reformulated the sentence. "Equation (2.2) will overestimate CO consumption at higher temperature because in reality CO consumption will decrease at higher temperatures than optimum temperature, while $f_2$ will keep increasing with rising temperature." See line 171 to 174.**

25) - line 193: i do not understand what Pr(t,i) is

**Response: We have added the detailed description: "production rate at temperature $T(t, i)$ divided by production rate at reference temperature". $P_r(t, i)$ is also descripted in equation (3.1) and the paragraph right below. See line 181 to line 182 and line 191 to line 192.**

26) - line 198: should it be 20 cm SOC?

**Response: Corrected to "30 cm". See line 186.**

27) - lines 275 – 277: give some details on the scenarios and datasets

**Response: We have added details of meaning of RCP2.6, 4.5 and 8.5. "RCP2.6, 4.5 and 8.5 datasets are future climate projections with anthropogenic greenhouse gas emission radiative forcing of 2.6 W m$^{-2}$, 4.5 W m$^{-2}$ and 8.5 W m$^{-2}$, respectively, by 2100." See line 278 to line 280.**

28) line 297: is the reported correlation really *r*, or is it *r^2*? Also, a correlation coefficient *r* of 0.5 is usually not considered high correlation.

**Response: Correlation coefficient is reported in R. We have rewritten the sentence to give proper description now. See section 3.1, line 303 to line 304.**

29) - lines 299 - 300: please compare the RMSEs reported to the CO fluxes, in order to give an impression on the relative errors.

**Response: We have presented CO net flux rates in order to compare with RMSEs. Please find in Section 3.1, line 305.**

30) - lines 318 – 319: "consume 42% and 58% of the total consumption, and produce 41% and 59% of total production" – please reformulate

**Response: We have changed Section 3.2 only for the period 2000-2013. We have also fixed this by reformulating. "The Southern and Northern Hemispheres have 41% and 59% of the total consumption, and 47% and 53% of the total production, respectively (Table 3)". See line 321 to 323.**

31) - line 328: Table 3 does not show the deposition flux in mg/m2 day

**Response: We have changed section 3.2 only for the period 2000-2013. The table has been updated to only have CO flux values for the period 2000-2013 with MOPITT CO surface concentration data and CO deposition velocity for the period 1901-2013 with constant in time, spatially distributed CO surface concentration. We have added the description on how to get these numbers. "calculated by flux values divided by area". See section3.2, line 336 to line337**

32) - lines 329 – 343: I find the text in this paragraph somewhat misleading. The fluxes are presented as changing or increasing relative to the simulation for 20th century, which suggests a temporal evolution, but in fact they are different mainly because of a different model setup, i.e different atmospheric CO concentrations. Consider using "different" and "larger" instead of "increasing".

**Response: We have removed this part and only talked about simulation results during 2000-2013 in section 3.2. We moved 1901-2013 deposition velocity simulation results to Section 3.3.**

33) - line 359: "the rate ranges of increasing of consumption…" – I think it should be something like "the ranges of the rates of increase in consumption…"; please explain what these ranges are, are they corresponding to the three scenarios?

**Response: We have changed the results in Section 3.3 to present deposition velocity and added the results of 1901-2013. We have rewritten this sentence and added more information to explain what the ranges are. See section 3.3, line 341 to line 343.**

34) - line 382: the references should be given in the method section, not here

**Response: Removed the references here. See section 4.1, line 390.**

35) - Sect 4.2: The information here belongs mostly to Results, please reorganize. Also, the section is hard to follow – there are many correlations mentioned without much coherence. Consider reformulating into a more focused way, with one paragraph per idea (e.g on annual time scales, the CO uptake is mostly correlated to X, Y, Z… and then comment more if needed on X ). Try to separate the annual and monthly results.

**Response: We have 1) moved the sensitivity results to new Section 3.4; 2) reorganized the rest part by totally separating the monthly and annual correlation test results. See Section 3.4, line 362 to line 369 and Section 4.2, line 409 to line 430.**

36) - lines 398 – 400 and Table 5: The effect of the SOC on the gross uptake flux seems too large. In my understanding, the text tries to explain that SOC increases the gross production which makes more CO available, which in turn leads to an increase in the gross CO consumption. But, for an increase in SOC of 30%, the production increases by about 10Tg/year, and the consumption increases by 28 Tg/year! How can that be? Where are the extra 18 Tg/year coming from?

**Response: We have added extra comments on this problem in discussion section 4.2. "To be noticed, the CO oxidation increasing due to CO substrate change is larger than production increasing due to SOC increasing, leading to an extra 18 Tg CO yr$^{-1}$ being taken up from the atmosphere to soils in sensitivity test when SOC increasing by 30% (Table 5)." See line 403 to 406.**

37) line 406: "as CO flux" – do you mean "as does the CO flux"?

**Response: This sentence has been removed now.**

38) line 415: what is 0.91? Is it r or r^2, or something else? The same for line 418.

**Response: We have added description. They are R. See section 4.2, line 424 and line 426.**

39) - line 425: the same data limitation is when using any method, not only SCE-UA-R, correct? If yes, remove "using SCE-UA-R method"

**Response: Yes. The data limitation is same to any method. We removed the "using SCE-UA-R method".**

**See section 4.3, line 434 to line 437.**

40) - line 427: "with RMSE … day-1" – I think this info has no meaning here

**Response: We have removed "with RMSE … day$^{-1}$" now. See section 4.3, line 437.**

41) - Table 6: what are the numbers? what are the units? These are not absolute values of fluxes.

**Response: They are correlation coefficient (R) between forcing variables and model results. We have**

**rewritten the description of Table 6 for better understanding. See table 6.**

42) - line 796: I suggest to replace "would happen inside" by "take place"

**Response: We have changed to "take place". See figure 1 caption, line 810.**

43) - Fig. 2: is this really volumetric soil moisture? The values do not seem realistic; I think typical values

for water holding capacity for most soils are around 50 % and that would give the saturation. Please

check the units.

**Response: Thank you for pointing this out. In this revision, we traced back to Nakai et al. (2013) and**

**found that our units and values are the same as they presented. The reason why the values were so**

**high is that the volumetric soil moisture (VSM) was converted from the water content reflectometry**

**(WCR) probe output period using an empirical calibration function of Bourgeau-Chavez et al. (2012)**

**for 5cm-30cm layer.  Although Bourgeau-Chavez et al. (2012) provided calibration functions for each**

**soil horizon (i.e., dead moss, upper duff, lower duff, and mineral soil), some of them resulted in values**

**greater than 100% VSM in Nakai et al. (2013) study. The model estimated high VSM (close to 80%) is**

**due to top 10 cm moss in the model which has a saturation VSM of 0.8.   We added the discussion on**

**Figure 2 caption in this revision. From line 482 to 485.**

44) Fig. 2-d2: please use the same units as in a2, b2 and c2 figures.

**Response: We have corrected units for figure 2d2.**

**Text comments:**

45) - line 60: should be "… consumption to be …"

**Response: Corrected. See line 53.**

46) - line 81: "formations" should be "formation"

**Response: Corrected. See line 63.**

47) - line 118: "are in Pihlatie" should be "is Pihlatie"

**Response: Corrected. See line 95.**

48) - line 219: I think "misfit" should be "mismatch"

**Response: We have changed to "mismatch". See line 218.**

49) - line 266 and through the paper: "transient" is used wrongly. If what is meant is "variable" then please do use "variable". Transient does not mean variable, but something that disappears.

**Response: We want to use "transient" with meaning of data having time variation. But we have removed all of them now in manuscript to avoid misleading.**

50) - Fig. 3: axes text too small; markers not visible in all figures (especially c2) , please consider using color markers

**Response: We have enlarged the axes text and used green diamond markers. Please see figure 3.**

51) - Fig. 3 part 1, page 34: Remove from caption the explanation for c and d

- Fig. 3 part 2, page 35: Remove from caption the explanation for a and b

**Response: We have removed them. Now figure 3 should have right description. See figure 3**

52) - Fig. 6: x labels not visible; some of the y labels cut

**Response: We have remade the figure and now the x and y labels can all be seen. See figure 6**

53) - Fig. 8: typo in legend: "producion"
- Fig. 9: typo in legend: "producion"

**Response: We have remade the figure 8 and figure 9. Now they have "production" in the legend. See figure 8 and 9.**

---

## Referee Report (RR1)

**Review of**

**Global Soil Consumption of Atmospheric Carbon Monoxide…**

by Liu et al., 2018

I find the revised version of the paper significantly improved. I have few comments, mostly technical. I recommend publication after these are addressed.

**Main comment - lines 403 – 406  and Table 5**

The effect of the SOC on the gross uptake flux seems too large (Table 5). A note confirming this has been added in this version of the manuscript (lines 403 – 406) but there is still no explanation given.

The 30% extra SOC increases the gross production, which makes more CO available in the soil, which in turn leads to an increase in the gross CO consumption from the soil. But, for an increase in SOC of 30%, the production increases by about 10Tg/year, and the gross consumption increases by 28 Tg/year! This means that an extra 18 Tg /year is taken from the atmosphere. I do not understand how this happens, as I think an increase in soil CO concentration cannot lead to an increase in the uptake of CO from atmosphere. (An increase of uptake from the atmosphere would actually need a decrease in the CO concentration in the upper layer of soil.) Also, from Eq. 2.x, it is clear that the uptake is related to SOC only via the CO soil concentration.

Please explain and/or correct if this is an error.

**Technical and text comments**

line 23: CO deposition velocity – net or gross?

line 25: why will deposition velocity increase? which are the main factors? Add a short explanation, e.g. "mainly due to xxx" xxx = increase in temperature?

line 54: since you just shown negative values, I think it is belter to specify here what are the negative fluxes, e.g.  (negative values represent deposition from atmosphere to soil)

lines 55 – 56: "All existing estimates have large  uncertainties ranging from -16 to -640 Tg CO yr-1" is this the range for estimates, or for uncertainties? – unclear, please reformulate

lines 56 – 57: "the estimates of CO dry deposition velocities also have large uncertainties, ranging from 0 to 4.0mm s–1" – is this the range for estimates, or for uncertainties? – unclear, please reformulate

lines 145 – 147: I think a piece of text is missing here. D is calculated using the method of Potter. Equations 2 – 4 are not related to D, but to the production and consumption rates P and O. Please check.

from line 155 till the end of this section: it is unclear what some of the numbers represent. E.g. why Vmax is a range and not one value? It would be good to explain here shortly that Vmax (also other parameters) are ecosystem specific, and that's why there are multiple values. This is shown later in Sect 2.3 but not clear here.

Also, are the Vmax values taken from Whalen & Reeburgh, 2001, or optimized in this work? If they are optimized in this work (as described in Sect. 2.3) then remove the reference to Whalen & Reeburgh (line 156). The same for kCO, and all the parameters optimized in this work – the way some of the references are given now suggests that the values are taken from those papers.

lines 210 – 211: "uptakes", "depositions" and "emissions" should be "uptake", "deposition" and "emission"

line 260: I suggest to state here explicitly that the CO surface concentration is constant in time.

lines 314 – 315: I think the ranges should be given with the lower value first, e.g. "-180 to -197" should be "-197 to -180"; same for -145 to -163. Same for the following: abstract;  lines 373 – 374, 483

lines 343 – 345: Fig. 9 does not show consumption, production and net flux

line 351: replace "from RCP2.6 to 8.5" by "from RCP2.6 to RCP8.5"

line 355: Table 5 should be Table 4

lines 450 – 452: What is meant here by "derived CO concentration"? – is it the constant, latitude dependent CO derived with the function 5? Please clarify. Also, if this is the case, a lower than real CO concentration will lead to an underestimation of CO deposition, and not to an overestimation, correct?

Table 5 – please check the sign of the % values. I think for Consumption and Net flux all the % values have the wrong sign. E.g. the change in consumption from baseline to (CO + 30% ) should be + 11% (-164.14 - (-147.65) / (-147.65) = 11.17).

Fig. 3.  I find it difficult to see some of the plots, in particular the measurement data in plots a2 and c2. Also the x-axis labels of a1, b1, c1 and d1 are difficult to see at normal page zoom.

Fig. 9 caption: remove the first "Future"

Fig. 12. typo in all x-axes labels, should be "concentration"

---

## Author Response (AR2)

**Author's response to referee 1:**

Thank you very much for your constructive comments!  You helped us improve this study significantly.

1) Main Comments: lines 403 – 406 and Table 5 The effect of the SOC on the gross uptake flux seems too large (Table 5). A note confirming this has been added in this version of the manuscript (lines 403 – 406) but there is still no explanation given. The 30% extra SOC increases the gross production, which makes more CO available in the soil, which in turn leads to an increase in the gross CO consumption from the soil. But, for an increase in SOC of 30%, the production increases by about 10Tg/year, and the gross consumption increases by 28 Tg/year! This means that an extra 18 Tg /year is taken from the atmosphere. I do not understand how this happens, as I think an increase in soil CO concentration cannot lead to an increase in the uptake of CO from atmosphere. (An increase of uptake from the atmosphere would actually need a decrease in the CO concentration in the upper layer of soil.) Also, from Eq. 2.x, it is clear that the uptake is related to SOC only via the CO soil concentration. Please explain and/or correct if this is an error.

**Response: Thank you for pointing this out. We have double checked the sensitivity results and found the extra 18 Tg CO yr$^{-1}$ from air to soils is due to current time step (5min) in combination with sudden a 30% SOC increase.  To explain this, we have tested the model using different time steps and SOC increasing percentages.  In this revision, we have added a new table (Table 7) to show the test results and  discussed this issue in Section 4.3 and lines 452 to 462: "**Fourth, from sensitivity test (Table 5) and model test (Table 7), we notice that the diffusion and consumption in the model is very sensitive to sudden 30% SOC changes with 5-minute time step. In reality, diffusion and consumption shall only be slightly influenced by indirect changes of soil CO concentration due to SOC changes. When we used 3-minute or 1-minute time step, the model responses to SOC changes are reasonable (Table 7). However, we believe 5-minute step is suitable in this study since SOC varies slightly during the whole global simulation period with only 3% increasing from 1900 to 2013 (Figure 4d) and up to a 4%

increase from 2014 to 2100 (Figure 6g).  Our model test showed there are small responses to these small amounts of SOC increasing (Table 7)."

**Technical and text comments**

2) line 23: CO deposition velocity – net or gross?

**Response: It is "net" deposition velocity. We have corrected in line 23.**

3) line 25: why will deposition velocity increase? which are the main factors? Add a short explanation, e.g. "mainly due to xxx" xxx = increase in temperature?

**Response: We have added explanation at the end in lines 24 to 27. "Under the future climate**

**scenarios, the CO deposition velocity will increase at 0.0002-0.0013 mm s$^{-1}$ yr$^{-1}$ during 2014-2100,**

**reaching 0.20-0.30 mm s$^{-1}$ by the end of the 21st century, primarily due to increasing temperature."**

4) line 54: since you just shown negative values, I think it is belter to specify here what are the negative fluxes, e.g. (negative values represent deposition from atmosphere to soil)

**Response: We have revised the sentence based on your suggestion in line 54. "negative values**

**represent the uptake from the atmosphere to soil"**

5) lines 55 – 56: "All existing estimates have large uncertainties ranging from -16 to -640 Tg CO yr-1" is this the range for estimates, or for uncertainties? – unclear, please reformulate

**Response: We have reformulated the sentence based on your comment in lines 55 to 56. "All existing**

**estimates have large uncertainties and range from -640 to -16 Tg CO yr$^{-1}$"**

6) lines 56 – 57: "the estimates of CO dry deposition velocities also have large uncertainties, ranging from 0 to 4.0mm s−1" – is this the range for estimates, or for uncertainties? – unclear, please reformulate

**Response: We have reformulated the sentence based on your comment in lines 57 to 58. "Similarly,**

**the estimates of CO dry deposition velocities also have large uncertainties and range from 0 to 4.0mm**

**s$^{-1}$"**

7) lines 145 – 147: I think a piece of text is missing here. D is calculated using the method of Potter.

Equations 2 – 4 are not related to D, but to the production and consumption rates P and O. Please check.

**Response: Thank you for pointing this out. This is a mistake. We have removed this from line 146.**

8) from line 155 till the end of this section: it is unclear what some of the numbers represent. E.g. why

Vmax is a range and not one value? It would be good to explain here shortly that Vmax (also other parameters) are ecosystem specific, and that's why there are multiple values. This is shown later in Sect

2.3 but not clear here.

Also, are the Vmax values taken from Whalen & Reeburgh, 2001, or optimized in this work? If they are optimized in this work (as described in Sect. 2.3) then remove the reference to Whalen & Reeburgh (line

156). The same for kCO, and all the parameters optimized in this work – the way some of the references are given now suggests that the values are taken from those papers.

**Response: Thank you for pointing this out. They are observed or estimated values from previous**

**studies, which will be used as a prior during our parameterization. We have added explanations for**

**this in lines 154 to 156. "Where $V_{max}$ is the ecosystem-specific maximum oxidation rate and was**

**estimated previously ranging from 0.3 to 11.1 µg CO g$^{-1}$ h$^{-1}$ for different ecosystems (Whalen &**

**Reeburgh, 2001)". And lines 165 to 166, "and their values were previously estimated ranging from 5 to**

**51 µl CO l$^{-1}$ for different ecosystems (Whalen & Reeburgh, 2001)"**

9) lines 210 – 211: "uptakes", "depositions" and "emissions" should be "uptake", "deposition" and

"emission"

**Response: We have corrected this. Please see lines 211 to 212. "Positive values of $v_d$ are soil uptake**

**(deposition from air to soils) and negative values are soil emissions."**

10) line 260: I suggest to state here explicitly that the CO surface concentration is constant in time.

**Response: We have revised the sentence based on your comment in lines 262 to 263. "to calculate the**

**distribution of static CO surface concentrations"**

11) lines 314 – 315: I think the ranges should be given with the lower value first, e.g. "-180 to -197"

should be "-197 to -180"; same for -145 to -163. Same for the following: abstract; lines 373 – 374, 483

**Response: Thank you for your suggestion. We have revised all places through the manuscript to use**

**values lower value first. Please see lines 16 to 17, line 53, line 56, lines 315 to 315, line 336, line 374 to**

**375, line 383, and lines 492.**

12) lines 343 – 345: Fig. 9 does not show consumption, production and net flux

**Response: Thank you for pointing this out. We have corrected this sentence in lines 344 to 345.**

**"During 2014-2100, there are significant trends of increasing deposition velocities for nearly all**

**scenarios (Figure 9)."**

13) line 351: replace "from RCP2.6 to 8.5" by "from RCP2.6 to RCP8.5"

**Response: We have corrected this in line 352. "Deposition velocities are increasing from RCP2.6 to**

**RCP8.5 and larger than in the historical periods in areas near the equator (Figure 10)."**

14) line 355: Table 5 should be Table 4

**Response: We have corrected this in lines 355 to 356. "Different vegetation types have a large range**

**of deposition velocity, from 0.008 to 1.154 mm s$^{-1}$ (Table 4)."**

15) lines 450 – 452: What is meant here by "derived CO concentration"? – is it the constant, latitude dependent CO derived with the function 5? Please clarify. Also, if this is the case, a lower than real CO

concentration will lead to an underestimation of CO deposition, and not to an overestimation, correct?

**Response: Thank you for pointing this out. We have rewritten the sentence and corrected to**

**"underestimation" in lines 450 to 452. "Third, the static CO surface concentration derived from the**

**empirical function is lower than MOPITT CO surface concentration, which will lead to underestimation**

**of CO deposition velocity during 1901-2100."**

16) Table 5 – please check the sign of the % values. I think for Consumption and Net flux all the % values have the wrong sign. E.g. the change in consumption from baseline to (CO + 30% ) should be + 11% (-

164.14 - (-147.65) / (-147.65) = 11.17).

**Response: Thank you for pointing out this mistake. We have made corrections in Table 5.**

17) Fig. 3. I find it difficult to see some of the plots, in particular the measurement data in plots a2 and c2. Also the x-axis labels of a1, b1, c1 and d1 are difficult to see at normal page zoom.

**Response: We have re-plotted the figure using two times bigger symbol for measurement data in**

**Figures 3a2, 3b2, 3c2 and 3d2. Also we have used two times bigger x-axis label for Figures 3a1, 3b1,**

**3c1, and 3d1. Please see Figure 3.**

18) Fig. 9 caption: remove the first "Future"

**Response: We have removed it.**

19) Fig. 12. typo in all x-axes labels, should be "concentration"

**Response: We have corrected the x-axis labels for figure 12.**

[revised manuscript text omitted]

velocity during 1901-2100. Fourth, f Fourth, the time step (5min) we used for CO
dynamics simulation is still not fine enough even we used Crank-Nicolson method to
reduce the influences. From sensitivity test ( in tTable 5) and model test (Tin table 7),
we can notice that 5 minute time step will let the diffusion and consumption in the model
is toevery sensitive to sudden 30% SOC changes with 5-minute time step. In reality,
diffusion and consumption shall , which should only be slightly influenced by indirect
changes of soil CO concentration due to SOC changes. When we used 3- minutes or 1-
minute time step, the model responses to SOC changes are increasing started to be
reasonable (Table 7). However, we believe Moreover, our model using 5- minute step is
is suitable in this study since SOC varies slightly during the whole global simulation
period with only (3% increasing from 1900 to 2013 ( Figure 4d) and ; up to a 4%
increase ing from 2014 to 2100 ( Figure 6g). Our , and model test showed there are
small tiny responses to these small amounts of is much SOC increasing (Table 7). A
finer time step (<1min) should be used to reduce the uncertainty in future works, if

[revised manuscript text omitted]

---

## Author Response (AR3)

**1 Author's response to co-editor:**

| 2                                                                                                          | Thank you very much for your comment! You helped us improve this study significantly.                                                                                                                                                                                                                                                                                                                                                                                                                                                                                                                                                                                                                                                                                                                                                                                                                                                                      |
|------------------------------------------------------------------------------------------------------------|------------------------------------------------------------------------------------------------------------------------------------------------------------------------------------------------------------------------------------------------------------------------------------------------------------------------------------------------------------------------------------------------------------------------------------------------------------------------------------------------------------------------------------------------------------------------------------------------------------------------------------------------------------------------------------------------------------------------------------------------------------------------------------------------------------------------------------------------------------------------------------------------------------------------------------------------------------|
| 3                                                                                                          | Comment: The referee asked why the (negative) deposition flux should INcrease when the SOC                                                                                                                                                                                                                                                                                                                                                                                                                                                                                                                                                                                                                                                                                                                                                                                                                                                                 |
| 4                                                                                                          | increases (table 5 and 7). This question is based on the assumption that higher SOC provides more                                                                                                                                                                                                                                                                                                                                                                                                                                                                                                                                                                                                                                                                                                                                                                                                                                                          |
| 5                                                                                                          | substrate for CO formation in the soil, which would DEcrease the soil-atmosphere gradient and DEcrease                                                                                                                                                                                                                                                                                                                                                                                                                                                                                                                                                                                                                                                                                                                                                                                                                                                     |
| 6                                                                                                          | the deposition flux. There must be one process in your model that leads to an increase in the deposition                                                                                                                                                                                                                                                                                                                                                                                                                                                                                                                                                                                                                                                                                                                                                                                                                                                   |
| 7                                                                                                          | flux.                                                                                                                                                                                                                                                                                                                                                                                                                                                                                                                                                                                                                                                                                                                                                                                                                                                                                                                                                      |
| 8                                                                                                          | You argue now with time step arguments, which apparently play a role, but even with a short timestep                                                                                                                                                                                                                                                                                                                                                                                                                                                                                                                                                                                                                                                                                                                                                                                                                                                       |
| 9                                                                                                          | the sign of the flux change remains the same, even if the magnitude is reduced. I think it is still required                                                                                                                                                                                                                                                                                                                                                                                                                                                                                                                                                                                                                                                                                                                                                                                                                                               |
| 10                                                                                                         | that you explain the physical basis of the effect. And then it would also be good if you can argue a bit                                                                                                                                                                                                                                                                                                                                                                                                                                                                                                                                                                                                                                                                                                                                                                                                                                                   |
| 11                                                                                                         | more quantitatively why the timestep has such a large influence (and that this then does not affect                                                                                                                                                                                                                                                                                                                                                                                                                                                                                                                                                                                                                                                                                                                                                                                                                                                        |
| 12                                                                                                         | other results of your model).                                                                                                                                                                                                                                                                                                                                                                                                                                                                                                                                                                                                                                                                                                                                                                                                                                                                                                                              |
|                                                                                                            |                                                                                                                                                                                                                                                                                                                                                                                                                                                                                                                                                                                                                                                                                                                                                                                                                                                                                                                                                            |
| 13                                                                                                         | Response: Thank you for your suggestions to improve the sensitivity analysis presented in the paper.                                                                                                                                                                                                                                                                                                                                                                                                                                                                                                                                                                                                                                                                                                                                                                                                                                                       |
| 13
14                                                                                                   | Response: Thank you for your suggestions to improve the sensitivity analysis presented in the paper.
In this revision, we removed Table 7 to avoid distraction from our main focus. We have also revised                                                                                                                                                                                                                                                                                                                                                                                                                                                                                                                                                                                                                                                                                                                                                |
| 13
14
15                                                                                             | Response: Thank you for your suggestions to improve the sensitivity analysis presented in the paper.
In this revision, we removed Table 7 to avoid distraction from our main focus. We have also revised
the sensitivity test (Table 5) by using SOC ±5% instead of ±30% (not a realistic variation), since during                                                                                                                                                                                                                                                                                                                                                                                                                                                                                                                                                                                                                                   |
| 13
14
15
16                                                                                       | Response: Thank you for your suggestions to improve the sensitivity analysis presented in the paper.
In this revision, we removed Table 7 to avoid distraction from our main focus. We have also revised
the sensitivity test (Table 5) by using SOC ±5% instead of ±30% (not a realistic variation), since during
our century-scale simulations, the SOC will not change beyond 4% within a century. This small                                                                                                                                                                                                                                                                                                                                                                                                                                                                                                                                  |
| 13
14
15
16
17                                                                                 | Response: Thank you for your suggestions to improve the sensitivity analysis presented in the paper.
In this revision, we removed Table 7 to avoid distraction from our main focus. We have also revised
the sensitivity test (Table 5) by using SOC ±5% instead of ±30% (not a realistic variation), since during
our century-scale simulations, the SOC will not change beyond 4% within a century. This small
variation of SOC did not affect CO consumption drastically, in contrast to the large effects due to                                                                                                                                                                                                                                                                                                                                                                                                                           |
| 13
14
15
16
17
18                                                                           | Response: Thank you for your suggestions to improve the sensitivity analysis presented in the paper.
In this revision, we removed Table 7 to avoid distraction from our main focus. We have also revised
the sensitivity test (Table 5) by using SOC ±5% instead of ±30% (not a realistic variation), since during
our century-scale simulations, the SOC will not change beyond 4% within a century. This small
variation of SOC did not affect CO consumption drastically, in contrast to the large effects due to
sudden high and unrealistic SOC changes in the original sensitivity test. Consequently, our other                                                                                                                                                                                                                                                                                                                      |
| 13

19                                                                     | Response: Thank you for your suggestions to improve the sensitivity analysis presented in the paper.
In this revision, we removed Table 7 to avoid distraction from our main focus. We have also revised
the sensitivity test (Table 5) by using SOC ±5% instead of ±30% (not a realistic variation), since during
our century-scale simulations, the SOC will not change beyond 4% within a century. This small
variation of SOC did not affect CO consumption drastically, in contrast to the large effects due to
sudden high and unrealistic SOC changes in the original sensitivity test. Consequently, our other
results of the global CO consumption simulations were not significantly affected by small variations of                                                                                                                                                                                                           |
| <ol> <li>13</li> <li>14</li> <li>15</li> <li>16</li> <li>17</li> <li>18</li> <li>19</li> <li>20</li> </ol> | Response: Thank you for your suggestions to improve the sensitivity analysis presented in the paper.
In this revision, we removed Table 7 to avoid distraction from our main focus. We have also revised
the sensitivity test (Table 5) by using SOC ±5% instead of ±30% (not a realistic variation), since during
our century-scale simulations, the SOC will not change beyond 4% within a century. This small
variation of SOC did not affect CO consumption drastically, in contrast to the large effects due to
sudden high and unrealistic SOC changes in the original sensitivity test. Consequently, our other
results of the global CO consumption simulations were not significantly affected by small variations of
SOC (less than 4%).                                                                                                                                                                                    |
| 13

21                                                         | Response: Thank you for your suggestions to improve the sensitivity analysis presented in the paper.
In this revision, we removed Table 7 to avoid distraction from our main focus. We have also revised
the sensitivity test (Table 5) by using SOC ±5% instead of ±30% (not a realistic variation), since during
our century-scale simulations, the SOC will not change beyond 4% within a century. This small
variation of SOC did not affect CO consumption drastically, in contrast to the large effects due to
sudden high and unrealistic SOC changes in the original sensitivity test. Consequently, our other
results of the global CO consumption simulations were not significantly affected by small variations of
SOC (less than 4%).
We followed your suggestion to mainly explain the physical basis of the effect in this revision. Please                                                                         |
| 13

22                                                   | Response: Thank you for your suggestions to improve the sensitivity analysis presented in the paper.In this revision, we removed Table 7 to avoid distraction from our main focus. We have also revisedthe sensitivity test (Table 5) by using SOC ±5% instead of ±30% (not a realistic variation), since duringour century-scale simulations, the SOC will not change beyond 4% within a century. This smallvariation of SOC did not affect CO consumption drastically, in contrast to the large effects due tosudden high and unrealistic SOC changes in the original sensitivity test. Consequently, our otherresults of the global CO consumption simulations were not significantly affected by small variations ofSOC (less than 4%).We followed your suggestion to mainly explain the physical basis of the effect in this revision. Pleasefind our changes for Table 5 and revisions on lines 126, 296, 364, 367, and 404-405 for sensitivity test |

[revised manuscript text omitted]
})$ | $\begin{array}{c} M_{opt} \\ (\frac{v}{v}) \end{array}$ | E SOC | $F_{SOC}$
$(rac{g}{g})$ | Ea ref
R
(K) | $\frac{PM_{ref}}{(\frac{v}{v})}$ | РТref
(°С) |
|----|-----------------------------------------|------------------------------------------------|-----------------------------|--------------------------------|-------------------|------------------------------|------------------------------|---------------------------------------------------------|------------------|-----------------------------|-------------------------------|----------------------------------|---------------------------------|
| 1  | Alpine Tundra & Polar Desert            | 36.00                                          | 0.78                        | 4.00                           | 1.80              | 0.10                         | 1.00                         | 0.55                                                    | 3.00             | 0.33                        | 7700                          | 0.25                             | 30.00                           |
| 2  | Wet Tundra                              | 36.00                                          | 0.70                        | 4.00                           | 1.80              | 0.25                         | 1.00                         | 0.55                                                    | 3.00             | 0.42                        | 7700                          | 0.25                             | 30.00                           |
| 3  | Boreal Forest                           | 27.34                                          | 1.18                        | 9.81                           | 1.60              | 0.15                         | 0.64                         | 0.53                                                    | 2.98             | 0.50                        | 8827                          | 0.35                             | 26.99                           |
| 4  | Temperate Coniferous Forest             | 42.64                                          | 2.15                        | 6.90                           | 1.87              | 0.02                         | 0.96                         | 0.53                                                    | 2.86             | 0.50                        | 8404                          | 0.38                             | 31.52                           |
| 5  | Temperate Deciduous Forest              | 40.16                                          | 2.43                        | 8.54                           | 1.51              | 0.17                         | 0.81                         | 0.51                                                    | 2.45             | 0.50                        | 8801                          | 0.35                             | 37.44                           |
| 6  | Grassland                               | 42.41                                          | 0.49                        | 11.27                          | 1.65              | 0.16                         | 0.82                         | 0.51                                                    | 3.09             | 0.42                        | 14165                         | 0.24                             | 12.29                           |
| 7  | Xeric Shrublands                        | 8.00                                           | 0.30                        | 4.00                           | 1.50              | 0.10                         | 1.00                         | 0.55                                                    | 3.00             | 0.33                        | 7700                          | 0.25                             | 30.00                           |
| 8  | Tropical Forest                         | 45.00                                          | 2.00                        | 4.00                           | 1.50              | 0.10                         | 1.00                         | 0.55                                                    | 3.80             | 0.50                        | 14000                         | 0.50                             | 18.00                           |
| 9  | Xeric Woodland                          | 8.00                                           | 0.30                        | 4.00                           | 1.50              | 0.10                         | 1.00                         | 0.55                                                    | 3.00             | 0.50                        | 7700                          | 0.25                             | 30.00                           |
| 10 | Temperate Evergreen
Broadleaf Forest | 40.16                                          | 2.43                        | 8.54                           | 1.51              | 0.17                         | 0.81                         | 0.51                                                    | 2.45             | 0.50                        | 8801                          | 0.35                             | 37.44                           |
| 11 | Mediterranean Shrubland                 | 45.00                                          | 1.50                        | 4.00                           | 1.50              | 0.10                         | 1.00                         | 0.55                                                    | 3.00             | 0.33                        | 7700                          | 0.25                             | 30.00                           |
| ** | Largest Potential Value                 | 51.00                                          | 11.1                        | 15.00                          | 2.00              | 0.30                         | 1.00                         | 0.60                                                    | 3.80             |                             | 15000                         | 0.60                             | 40.00                           |
|    |                                         |                                                |                             |                                |                   |                              |                              |                                                         |                  |                             |                               |                                  |                                 |

[revised manuscript text omitted]

85B

87 Table 7. Model test for site No.8 during 2002-2003. Time step is for solving equation (1). SOC increasing

871 represents the percentage of SOC increased in each test. Baseline is the simulation using original time step

872 and SOC input. Differences represent new simulation results minus baseline results.

| Time Step | SOC Increasing | <del>Units: mg m-2 yr-1</del> | Consumption | Production       | Diffusion          |
|-----------|----------------|-----------------------------------------------------|--------------------|------------------|--------------------|
| 5min      | <del>0%</del>  | Baseline                                     | <del>-1611.5</del> | <del>410.0</del> | <del>-1201.5</del> |
| 5min      | <del>30%</del> | Differences                                  | <del>-293.0</del>  | <del>123.0</del> | <del>-170.0</del>  |
| 3min      | <del>30%</del> | Differences                                  | <del>-156.7</del>  | <del>123.0</del> | <del>-33.7</del>   |
| 1min      | <del>30%</del> | Differences                                  | <del>-97.4</del>   | <del>123.0</del> | <del>25.6</del>    |
| 5min      | <del>10%</del> | Differences                                  | <del>-97.7</del>   | <del>41.0</del>  | <del>-56.7</del>   |
| 5min      | <del>1%</del>  | Differences                                         | <del>-9.8</del>    | 4.1              | <del>-5.7</del>    |